# Global analysis of ocean phytoplankton nutrient limitation reveals high prevalence of co-limitation

Thomas J. Browning [1] ✉ & C. Mark Moore[2] ✉

Nutrient availability limits phytoplankton growth throughout much of the global ocean. Here we synthesize available experimental data to identify three dominant nutrient limitation regimes: nitrogen is limiting in the stratified subtropical gyres and in the summertime Arctic Ocean, iron is most commonly limiting in upwelling regions, and both nutrients are frequently co-limiting in regions in between the nitrogen and iron limited systems. Manganese can be co-limiting with iron in parts of the Southern Ocean, whilst phosphate and cobalt can be co-/serially limiting in some settings. Overall, an analysis of experimental responses showed that phytoplankton net growth can be significantly enhanced through increasing the number of different nutrients supplied, regardless of latitude, temperature, or trophic status, implying surface seawaters are often approaching nutrient co-limitation. Assessments of nutrient deficiency based on seawater nutrient concentrations and nutrient stress diagnosed via molecular biomarkers showed good agreement with experimentally-assessed nutrient limitation, validating conceptual and theoretical links between nutrient stoichiometry and microbial ecophysiology.

The growth of marine phytoplankton is commonly limited by the availability of one or more nutrients[1]. Knowledge of the identity of these nutrients, and how their external supply impacts phytoplankton abundance and activity, are crucial for understanding and predicting the marine ecosystem responses to altered nutrient supply to the surface ocean, which may be associated with past and ongoing environmental changes[1–5]. Such knowledge is subsequently key for the Earth System as a whole and carries strong economic and humanitarian importance, as phytoplankton activity regulates global nutrient cycles, atmosphere-ocean carbon exchange and the amount of carbon fixed and energy made available to higher trophic levels[5]. Understanding of nutrient limitation patterns is also important for rigorous assessment of Earth System Models, which still often disagree on the identity of limiting nutrients, at least at regional scales, potentially contributing to uncertainties in phytoplankton responses to climate change[4,5].

Establishing which nutrient is growth-limiting to phytoplankton has most commonly used an experimental approach: seawater is amended with the nutrient(s) hypothesized to be limiting and incubated for a set period of time before assessing phytoplankton biomass changes relative to initial conditions and/or untreated controls[6,7]. Assessments of significant positive enhancements in phytoplankton biomass following nutrient amendment relative to controls are interpreted to reflect an in-situ condition of limitation by the added nutrient(s). Implicit in this assessment are that (i) only the intended resource is supplied and that any contamination by elements or changes in available light from the in-situ condition do not have a significant influence on the result (for example, conducting experiments without the interference of highly contamination-prone elements such as Zn), (ii) any differential changes in phytoplankton loss processes (grazing, viral lysis, mortality) between treatments over the experimental duration are smaller in magnitude than the nutrient-stimulated increase in growth[6,8].

Whilst in practice testing these assumptions is difficult, the use of stringent protocols to prevent nutrient contamination, coupled with a

[1]Marine Biogeochemistry Division, GEOMAR Helmholtz Centre for Ocean Research, Kiel 24148, Germany. [2]School of Ocean and Earth Science, National Oceanography Centre Southampton, University of Southampton, Southampton SO14 3ZH, UK. ✉e-mail: tbrowning@geomar.de; c.moore@noc.soton.ac.uk

lack of robust alternate approaches, has led to the relatively wide-spread use of such experimental assessments across the ocean[1]. Furthermore, when multiple nutrients are supplied in factorial combinations, these experiments have the capacity to reveal both the primary limiting nutrient and potential co-limiting nutrients; with co-limitation reflecting a situation where two or more nutrients simultaneously restrict phytoplankton growth[1,8–11]. Multiple different forms of co-limitation exist and differentiating between them using assessments of biomass changes, which provides an integrated response to shifts in multiple biochemical processes occurring from the cell to community level, is challenging[8,10,12]. However despite these caveats, both existing experimental compilations[1] and more recent experimental programmes[11,13] have demonstrated that, at the level of bulk phytoplankton biomass, independent co-limitation by two nutrients (where both nutrients need to be added to generate any biomass enhancement) and serial limitation (where addition of one nutrient leads to an enhancement in biomass, but adding an additional second or third nutrient leads to subsequently enhanced responses) are both readily resolvable and are indeed potentially widespread in the ocean.

An earlier synthesis of experimental evidence for nutrient limitation revealed a broad-scale pattern of N limitation in the stratified, low latitude subtropical gyres and Fe limitation in regions of oceanic upwelling associated with elevated N concentrations[1]. Despite indications in earlier synthesis studies[14,15], P was not found to be the primary limiting nutrient in any of the experiments in the compilation, although co-/serial P limitation was apparent in some regions[16–18]. Here we build on this earlier experimental compilation[1] by adding more recent experimental data, resulting in an approximate doubling of the number of experiments in the dataset. We also extract associated metadata from the original studies to enable a more complete quantitative analysis of experimental results. Our primary goals were to: (i) add to the spatial extent and resolution of experimentally-determined nutrient limitation in the global ocean; (ii) place recent findings of both co-limitation and limitation by nutrients other than N, P, and Fe into a global context; (iii) quantitatively evaluate differential phytoplankton growth responses to nutrient supply and dissect potential driving factors; (iv) evaluate qualitative and quantitative responses to nutrient treatments in the context of ambient seawater nutrient concentrations, and (v) compare meta-analysis of experimental data to recent molecular biomarker datasets of nutrient stress. Our analysis demonstrates that phytoplankton net growth is often significantly enhanced through increasing the number of different nutrients supplied, regardless of latitude, temperature, or trophic status, implying surface seawaters are often approaching a state of nutrient co-limitation.

## Results and discussion
### Patterns of oceanic nutrient limitation
Three main nutrient limitation provinces emerge from the compiled dataset: primary N and Fe limitation (39 and 32% of experiments; $n = 62$ and $n = 50$, respectively) and N-Fe co-limitation (9% of experiments, $n = 14$). The dataset is consistent with earlier reports in demonstrating widespread N limitation in the subtropical gyres, where surface N concentrations are depleted due to strong stratification of near-surface waters, and Fe limitation in the upwelling regions away from strong aerosol Fe sources, where N concentrations are elevated and Fe is often at low levels (Fig. 1)[1]. Primary Fe limitation in the latter was furthermore supported by the chlorophyll and/or primary production increases observed in ten, kilometre-scale in situ Fe enrichment experiments, which are also included in the dataset (Fig. 1). The updated database further includes the recent direct evidence for N limitation in the summertime Arctic Ocean[19,20], where deeper water N supply to stratified surface waters is also restricted[21] and Fe supply from terrestrial material is generally elevated[22]. Co-limitation by both N and Fe is frequently found in between the regions of N limitation and Fe limitation (Fig. 1)[11,13,23–28]. Furthermore, along the gradients between regions of primary N, primary Fe, and N-Fe co-limitation, serial responses to N or Fe are observed, demonstrating a coherent trend in experimental responses on transitioning across these three nutrient limitation provinces[11,13,23–28].

Consistent with more recent assessment[1] and in contrast to earlier syntheses[14,15], primary P limitation is not observed in any experiment in the dataset, although co-/serial P limitation was observed, particularly in northern hemisphere (sub-)tropical waters of the Mediterranean, North Atlantic subtropical gyre and eastern North Pacific subtropical gyre (Fig. 1). All these areas are

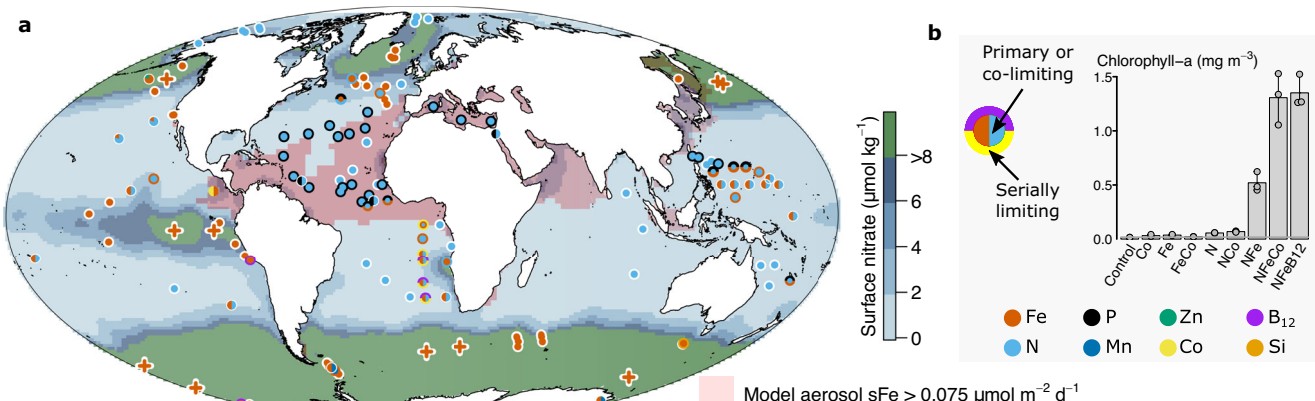

**Fig. 1 | Global synthesis of nutrient limitation. a** Experimental locations presented on a global map as coloured symbols. **b** Example experiment[11]. Legend next to example experiment indicates the identities of (co-)limiting nutrient(s) in (**a**) The central symbol colour(s) on the map indicate the primary limiting nutrient (i.e., adding this nutrient alone stimulated chlorophyll-a accumulation). Outer symbol colours (i.e., colours of the annulus) indicate serial limiting nutrient(s) (i.e., adding this nutrient in addition to the primary limiting nutrient(s) stimulated further growth than supplying the primary limiting nutrient(s) alone). Split colours for inner or outer symbol indicate nutrients that were co-limiting. Sequential levels of serial limitation are indicated by multiple layers of annuli, referencing to secondary limitation (inner annulus) and tertiary limitation (outer annulus). Co-limitation can either be at the primary (split central circle) or serial (split annulus) level. Mesoscale Fe enrichment experiments are shown as crosses. Background colours on map in (**a**) indicate annual average surface nitrate concentrations. Regions of elevated soluble aerosol Fe deposition predicted by a model are highlighted[77]. Bars in (**b**) represent the mean chlorophyll-a response to nutrient combinations after 48 h ($n = 3$ biologically independent samples), dots represent the responses of individual treatments, and the error bars indicate the range[11]. Source data are provided in Supplementary Data 1.

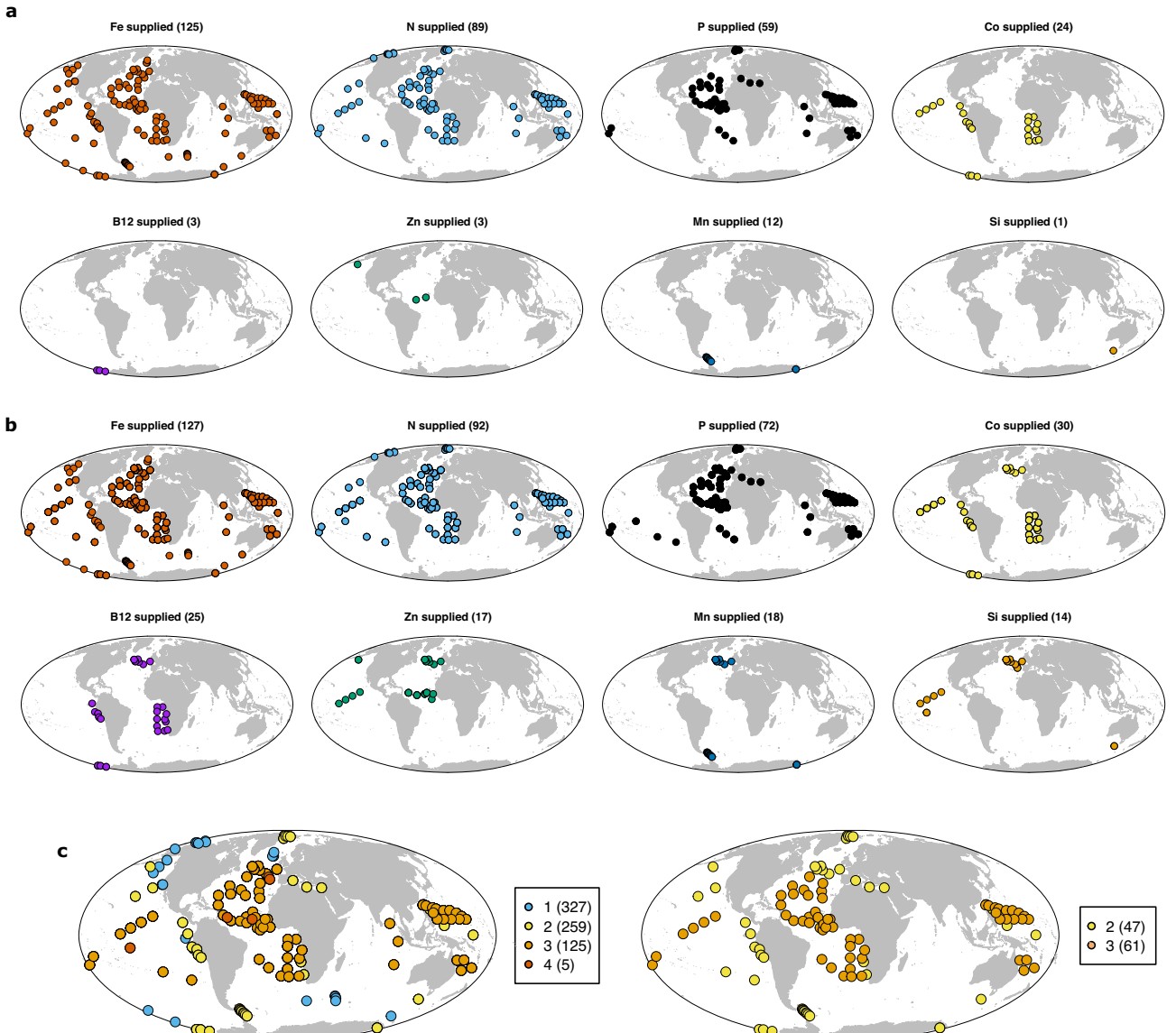

**Fig. 2 | Nutrients added within experimental nutrient additions.** Experiments where: (**a**) a given nutrient was supplied alone; (**b**) a given nutrient was supplied in any combination. **c** Maximum number of added nutrients in a given experiment. **d** Experiments where nutrients were added in full factorial combinations for 2 or 3 nutrients. Experimental and/or individual treatment numbers are indicated in brackets. Source data are provided in Supplementary Data 1.

characterised by restricted subsurface N and P supply combined with significant atmospheric Fe supply (Fig. 1). As previously discussed[1], the development of co-/serial P limitation in such systems is consistent with theoretical considerations, whereby relief of any Fe limitation of diazotrophs (N₂ fixing organisms) in low N environments allows them to remove residual P down to concentrations where community level N and P co-limitation can occur[29,30]. The only nutrient other than N or Fe in the dataset that was found to be primary limiting was Mn in the Southern Ocean[31–34]. The occurrence of primary Mn limitation, alongside evidence of co-/serial Fe-Mn limitation, in parts of the Southern Ocean is a result of Mn deficient deep waters upwelling to the surface ocean in regions where additional Mn sources (aerosols, sediments) are restricted[32,35–37]. Alongside the nutrients P and Mn, experiments performing individual additions of Co are relatively widespread in the dataset (*n* = 24; Fig. 2a). Whilst none of these experiments demonstrated primary Co limitation, Fe-Co co-limitation was observed at one site in the Costa Rica upwelling dome[38]. Furthermore, observations across two studies have found that following

relief of N-Fe co-limitation, serial (in this case tertiary level) Co limitation can be widespread throughout the upwelling-subtropical gyre boundary of the Southeast Atlantic[1,11].

There are insufficient tests to draw any robust conclusions with regards to primary limitation by nutrients other than N, P, Fe, Mn, or Co (Fig. 2a). For example, whilst both phytoplankton requirements and the potential for surface ocean depletion of silicic acid, Zn, and vitamin B₁₂ are well known, there are only 1, 3, and 3 experiments respectively in the dataset that tested for primary limitation by each of these nutrients (Fig. 2a). A greater number of experiments have been conducted with these elements in a serial addition scenario (Fig. 2b); that is, to assess if they are co- or serially limiting alongside or following supply of N, P, and/or Fe. The outcomes of such tests are varied. Silicic acid and vitamin B₁₂ can be serially limiting in the Southern Ocean (silicic acid, ref. 39 B₁₂, ref. 40) and South Atlantic (B₁₂, ref. 11), whilst Zn additions in the high latitude North Pacific generated an independent net growth response to a separate Fe additions[41]. Further experimental tests supplying these nutrients alone and in combination with the

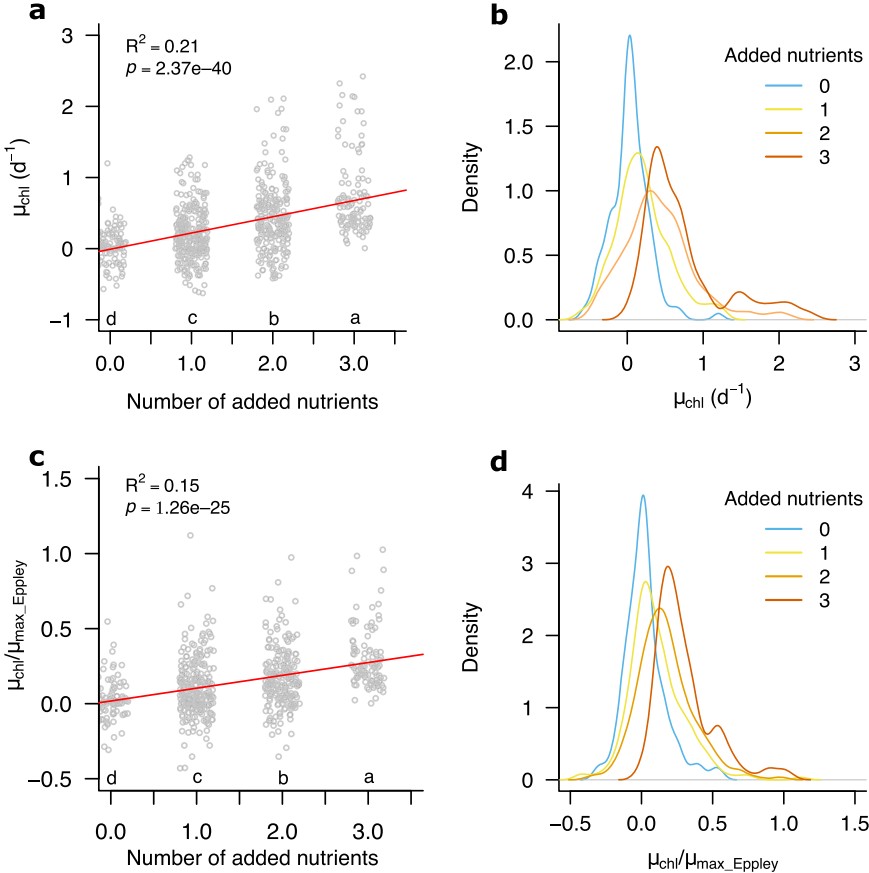

**Fig. 3 | Impact of multiple nutrient addition on estimated net phytoplankton growth. a, b** Without temperature normalization. **c, d** With temperature-based maximum growth rate normalization (see Methods Eq. 2 and Fig. 4b). Different letters below clusters in (**a, c**) indicate significantly different means (one-way ANOVA $p < 0.05$, followed by Tukey Honest Significant Difference test). In (**b, d**) density refers to the kernel density estimate[78]. Source data are provided in Supplementary Data 1.

nutrients likely to be primary limiting (N, Fe, and/or Mn) are thus still needed to draw more robust conclusions about the limitation potential for these nutrients. More generally, whilst co-/serial limitation is emerging as potentially widespread[11,13,26,31,32,38], there remain broad regions of the ocean where experimental tests for co-limitation have not been conducted (Fig. 2c, d); for example, the majority of experiments conducted so far in the Southern Ocean only supplied one nutrient (Fe).

**Multiple nutrient limitation**

Across the dataset as a whole, the additive supply of increasing numbers of different nutrients led to sequentially higher net chlorophyll-a growth rates ($R^2 = 0.21$, $p < 0.0001$, $n = 765$; Fig. 3). The mean net chlorophyll-a growth response was 0.03, 0.20, 0.42, and 0.73 d$^{-1}$ for addition of 0 (no nutrient control), 1, 2, and 3 nutrients respectively (Fig. 3a, b). This trend of increasing mean growth responses was accompanied by an increasing spread of responses (standard deviations of 0.25, 0.36, 0.47, and 0.54 d$^{-1}$ for 0, 1, 2, and 3 nutrients respectively). Part of the trend of increasing growth response with increasing numbers of nutrients added was potentially an artifact of temperature-associated growth rates, as when these rates were normalized to estimated maxima based on ambient temperature (see Fig. 4b)[42], the trend remained positive but became less strong ($R^2 = 0.15$; $p < 0.0001$, $n = 680$; Fig. 3c, d). The stronger non-temperature-normalized trend presumably therefore resulted at least partly from the assembled dataset containing more multi-nutrient treatments for those experiments performed in warmer, lower latitude waters (Figs. 2c, 4a, b).

Within any given system, multiple nutrient addition frequently resulted in greater net growth responses regardless of latitude, water temperatures, and chlorophyll-a (Fig. 4a–c). However, highest net chlorophyll-a growth rates following (multiple) nutrient supply were observed at lower latitudes, in warmer waters with lower initial chlorophyll-a concentrations (Fig. 4a–c). In general, therefore, a degree of multiple nutrient limitation (that is, co-limitation or serial limitation) appeared common regardless of the system, whilst absolute growth responses to nutrient supply appeared to be strongly modulated by the prevalent environmental and/or ecological conditions. Whilst latitude, temperature and initial chlorophyll-a could all be hypothesized to play a role in regulating the maximum growth response to supply of multiple nutrients, these environmental variables all co-vary across the dataset (latitude-temperature: $R^2 = 0.81$, $p < 0.0001$, $n = 117$; temperature-chlorophyll-a: $R^2 = 0.18$, $p < 0.0001$, $n = 112$; latitude-chlorophyll-a: $R^2 = 0.13$, $p < 0.0001$, $n = 119$). Correspondingly, when normalized to potential growth rate maxima predicted by ambient temperature, no clear trends were observed with latitude or chlorophyll-a concentrations (Fig. 4d–f). This suggested that, following addition of the limiting nutrient(s), temperature is potentially the main driver of the trends of absolute growth rate response (Fig. 4b), due to the expected temperature dependence of metabolic processes that subsequently regulates maximum, nutrient-replete growth rates[42–44] (see also Supplementary Text 1).

Although the temperature dependence of growth rate appeared to be the main driver of spatial patterns in response strength following addition of the limitation nutrient(s), other ecosystem factors may still play a role. For example, in the context

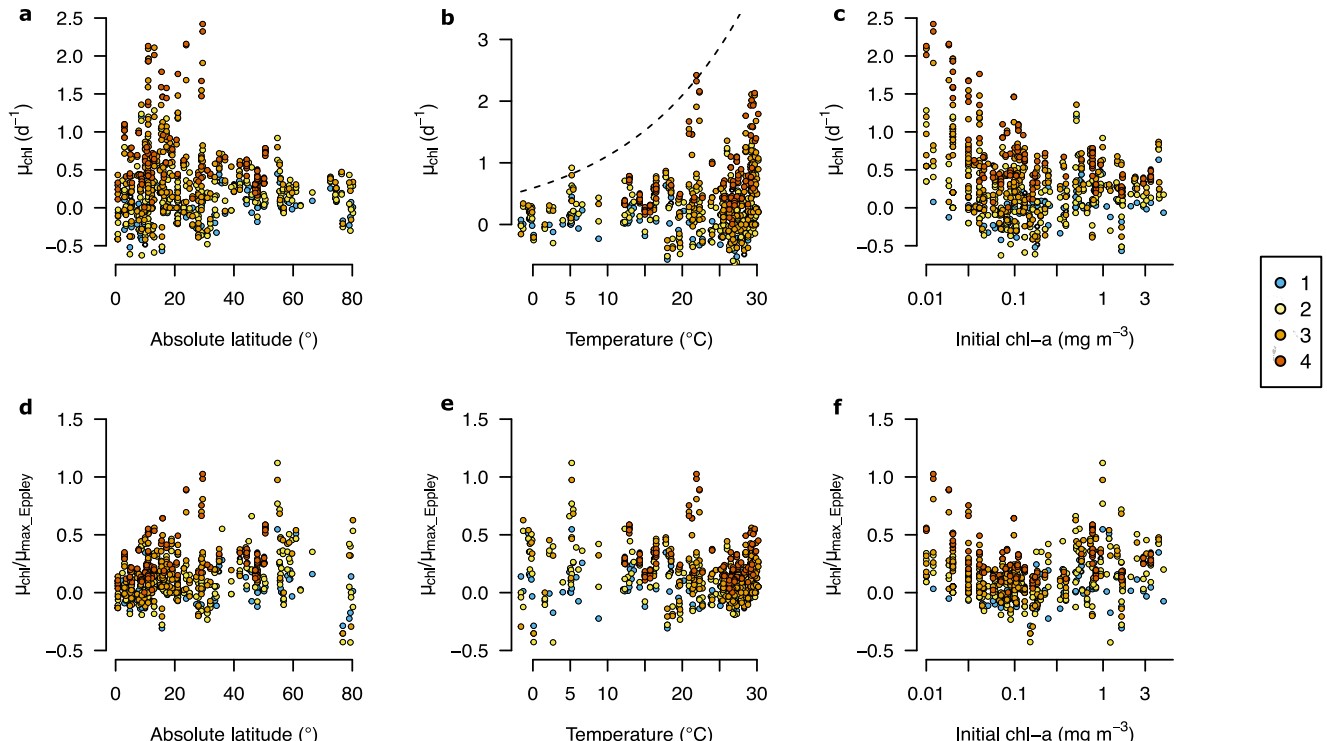

**Fig. 4 | Relationship of net chlorophyll-a growth rates following experimental nutrient addition with latitude, temperature and trophic status. a–c** Without temperature normalization. The dashed line in (**b**) is the estimated maximal growth rate based on an empirical relationship (Eq. 2)[42], which appears to define the upper envelope of responses reasonably well. **d–f** With temperature-based maximum growth rate normalization (see Methods). Symbol colours indicate numbers of added nutrients within individual experimental treatments. Source data are provided in Supplementary Data 1.

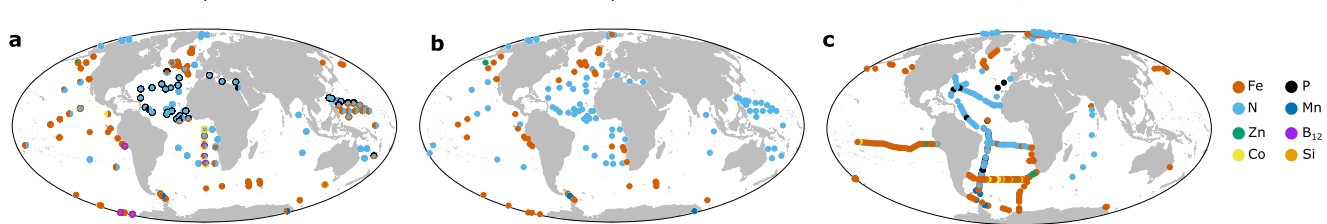

**Fig. 5 | Comparison of nutrient limitation and deficiency. a** Experimentally-determined nutrient limitation (see Fig. 1). **b** Prediction of most deficient nutrient at experimental sites based on seawater nutrient concentrations at the time of experimental water collection, combined with an assumed-average phytoplankton elemental stoichiometry (16 N: 1 P: $7.5 \times 10^{-3}$ Fe: $2.8 \times 10^{-3}$ Mn: $8 \times 10^{-4}$ Zn : $1.9 \times 10^{-4}$ Co; ref. 1 see Methods). **c** As for (**b**) but for a fuller range of nutrient elements from the GEOTRACES IDP2017 (V2)[50]. Colours correspond to added nutrients as indicated in the legend. Seawater concentrations of vitamin $B_{12}$ and silicic acid were not included in calculations as they were frequently not reported (vitamin $B_{12}$) and/or are required by only a few phytoplankton groups (silicic acid). Source data are provided in Supplementary Data 1 and via the publicly accessible GEOTRACES IDP2017 (V2)[50].

of such experiments, lower initial phytoplankton biomass levels will enable a greater number of divisions before one or more of the supplied nutrients are removed by biological uptake to once more approach growth-limiting levels. Moreover, within oligotrophic systems dominated by small celled phytoplankton, the zooplankton that graze on the potentially fastest responding, initially rarer, taxa may also be initially scarce, providing a greater window of opportunity for rapid phytoplankton growth responses[45]. Conversely, under higher chlorophyll-a conditions, grazers of bloom-forming phytoplankton are likely a more established part of the microbial community and can thus respond more rapidly and restrict overall net phytoplankton growth over the experimental duration[45]. The potentially larger sizes of such grazers may, however, also lead to their partial exclusion from the in vitro micro-/meso-cosm type enclosures which tended to dominate experimental designs within the dataset. However, previously observed consistencies between in vitro and in situ experimental responses, at least in some systems, argue against this being a major effect[46].

## Nutrient stoichiometry

Clear linkages were found between the nutrient found to be limiting experimentally and the nutrient predicted to be most deficient (Fig. 5)[1,36]. Making an assumption about phytoplankton requirements for different elements, dissolved seawater nutrient concentrations can be used to predict the nutrient that is most deficient in seawater[36]. The most deficient nutrient is subsequently predicted to be the nutrient limiting growth, or having the greatest potential to limit growth following continued phytoplankton growth and nutrient drawdown[1,36]. Calculating nutrient deficiency using the dissolved nutrient concentrations reported alongside each of the bioassay experiments in the dataset demonstrated a high level of predictability: 73.6% of the limiting nutrient predictions based on deficiency were correctly matched

by the experimental outcome (Fig. 5). In the remainder of the experiments, the deficiency approach was either incorrect (1 experiment, 0.7% of predictions) or could not generate an accurate prediction, as either (i) the concentration of the limiting nutrient was not determined (1 experiment, 0.7 % of predictions), or experiments demonstrated that (ii) no added nutrient was limiting (i.e., all were replete at the time of the experiment; 9.3% of predictions), or (iii) two nutrients, including that predicted to be most deficient, were co-limiting (15.7%). Situations where all nutrients are in excess ('ii') might be expected under one or a combination of light limitation, low temperatures, or strong grazing pressure[47,48]. As well as being relatively uncommon within the compiled data set, such situations are also potentially difficult to predict simply from dissolved nutrient concentrations. Lower concentration bounds for individual nutrients could be considered; however, in a natural system, the equilibrium nutrient concentration for a limiting nutrient is set by multiple ecosystem characteristics, including the grazing pressure and other constraints on growth[49]. Thus, the standing stocks of (co-)limiting nutrients can, at least in theory, vary considerably. Prediction of nutrient co-limitation (i.e., 'iii' above) from dissolved concentrations alone is similarly challenging. Once again, residual nutrient standing stocks following biological removal should provide an indicator of the transitions between regions of single nutrient limitation and co-limitation[11]. However, significant biological stoichiometric flexibility (including elemental substitutions[8]) potentially needs to be taken into account. However, it might be possible to set ranges in nutrient deficiency around the average stoichiometry where both nutrients are equally limiting[11].

Prediction of the results from the experimental dataset on the basis of independent dissolved nutrient deficiencies was also potentially complicated in a number of cases, where the concentrations of only a few nutrients were reported (typically N, P, and Fe). Additionally, the presence and potential biological use of frequently unreported dissolved nutrient pools (e.g., organic forms of N or P), represents a further complication. Nevertheless, the high apparent level of predictability of the experimental outcomes from measured dissolved nutrient concentrations still potentially provides a valuable approach for making wider predictions about nutrient limitation in regions where experiments have not been conducted, particularly where more complete datasets of nutrient concentrations are available[36]. For example, conducting the same calculations for samples from surface waters reported in the GEOTRACES intermediate data product[50], a dataset of accurate dissolved nutrient concentrations at sampling sites throughout the global ocean, broadly replicated spatial patterns observed from the bioassay dataset (Fig. 5c): N was predicted to be the most deficient and thus most likely to be limiting nutrient in the stratified low latitude gyres and the Arctic Ocean, whilst Fe was predicted to have highest limitation potential in upwelling regions. Superimposed on this broadscale trend were multiple sites in the Southern Ocean where Mn was predicted to be the most deficient, at the boundaries of the subtropical gyres where Co and Zn were predicted to be the most deficient, and in the (sub)tropical North Atlantic where P was predicted to be the most deficient. These patterns are broadly consistent with primary/serial/co-limitation (Mn) and serial (Co, P) limitation that have been experimentally determined in each of these respective settings (Fig. 1).

In addition to qualitative nutrient deficiency predictions of which nutrient was limiting, dissolved N:Fe ratios subsequently showed some capacity for quantitatively predicting net chlorophyll-a growth rates in experiments following addition of N or Fe, the nutrients most commonly found to be primary limiting in the dataset (Fig. 6a). At progressively increasing dissolved N:Fe, the magnitude of net growth following N addition decreased (for log10(N:Fe), $R^2 = 0.20$; $p = 0.00012$), whilst the magnitude of net growth following Fe addition increased (for log10(N:Fe), $R^2 = 0.24$; $p < 0.0001$). Net growth rates following combined N+Fe addition were frequently elevated above

both N and Fe additions throughout a broad range of N:Fe ratios. However, enhanced co-/serial N-Fe responses were less clear for experiments initiated at the highest dissolved N:Fe, which typically coincided with oceanic upwelling zones with high nitrate concentrations relative to those of Fe. Normalization to expected maximal growth rates based on ambient temperature led to similar trends except that responses to N addition were no longer statistically significantly related to increasing dissolved N:Fe (for log10(N:Fe), $R^2 = 0.03$, $p = 0.18$), while the relationship between dissolved N:Fe and responses to Fe addition became stronger (for log10(N:Fe), $R^2 = 0.33$, $p < 0.0001$). This was likely due to the prevalence of Fe limitation in colder, upwelling regions, whereas N limitation was found across a broader range of temperatures (e.g., (sub)tropical and Arctic).

In theory, the greatest enhancement of net growth rates following combined N+Fe additions relative to individual N and/or Fe additions should be observed at intermediate dissolved N:Fe ratios[10,11,51]. This intermediate dissolved N:Fe ratio should approximately correspond to the intersection point of the net growth rate–dissolved N:Fe ratio slopes for individual N and Fe additions and the value of typical phytoplankton N:Fe requirements, which appeared to match well with each other in our analysis (intersection of red, blue and dashed lines in Fig. 6a, c)[10,11,51]. Whilst previous results from individual systems appear to conform to experimental co-limitation responses being predictable on the basis of dissolved N:Fe ratios[11], several layers of complexity could and indeed do appear to complicate the situation within our larger combined dataset. Firstly, the N:Fe supply ratio, rather than standing N:Fe concentration ratio, would actually be expected to be the ultimate driver and therefore best predictor of (co-)limitation[51]. Predictability of net growth following N and Fe additions on the basis of residual concentration ratios (i.e., the concentrations which remain following biological uptake[11]) may thus be expected to have sensitivity to the particular system in question, including again through biological stoichiometric flexibility[1]. For instance, a set of experimental sites located in the western subtropical North Pacific showed a strong gradient from N limitation to N-Fe co-limitation across a region where surface water dissolved N:Fe concentration ratios remained low due to extremely depleted surface water N concentrations in all cases (nitrate<10 nM)[13]. The N-Fe co-limited locations observed in this study were, however, located over a much shallower nitracline due to wind-driven Ekman divergence, which was estimated to elevate N:Fe supply ratios and lead to the development of the observed co-limitation[13]. More broadly, experiments revealing N-Fe co-limitation typically correspond to regions with more elevated levels of sub-surface N supply than is reflected by the depleted surface concentrations (compare Fig. 7 with Fig. 1). Secondly, at very low concentrations of both N and Fe, the net growth that can be achieved following addition of one of the nutrients would be expected to be low regardless of their ratio in the initial seawater. Also, at such low concentrations, calculated residual surface water nutrient ratios will further be very sensitive to natural variability and measurement error associated with their quantification[52]. Thirdly, as indicated above, analysis on the basis of measured dissolved N and Fe again neglects the potential importance of nutrient speciation. For example, some of the measured trace metals might have limited bioavailability, while dissolved organic forms of nitrogen and phosphate were not considered, but might be bioavailable. Finally, as also mentioned above, the magnitude of net growth responses are under both ecological as well as environmental control, with the extant phytoplankton and their grazers within a system both regulating the magnitude of growth response[47,48].

In contrast to the case for N and Fe, which show clear gradients in the experimental response strengths between high N upwelling provinces and N depleted systems (Fig. 6a, c), initial experimental dissolved surface water N and P concentrations co-varied across the data set and the net growth responses to N and P addition showed a similar gradient across the range of seawater N:P concentration ratios (Fig. 6b,

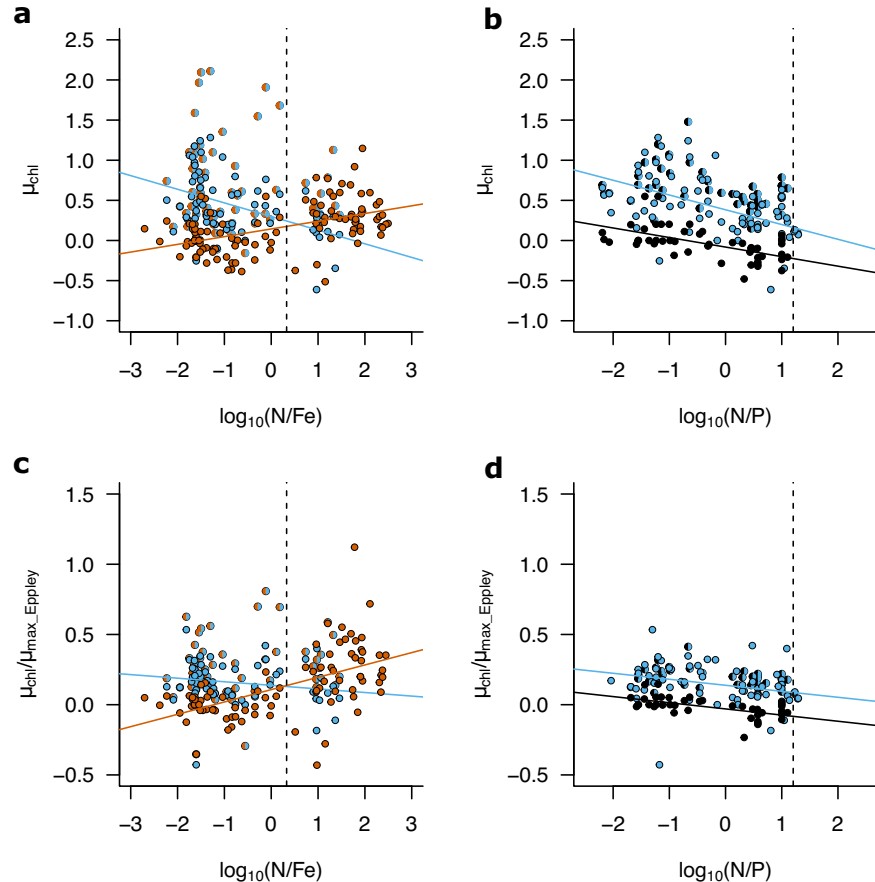

**Fig. 6 | Relationship of experimentally derived phytoplankton responses to nutrient supply with environmental dissolved nutrient concentration ratios. a, c** Net growth rates derived from changes in chlorophyll-a following N supply were higher at low N:Fe ratios and lower at high N:Fe ratios, with the reverse trend observed for growth rates resulting from Fe supply. Note that N:Fe is in units of mol: mmol. **b, d** N supply almost always led to higher net chlorophyll-a growth rates than

P supply throughout the range of encountered dissolved N:P ratios. Vertical dashed lines represent assumed-average phytoplankton N:Fe (2.13 mol:mmol) and N:P (16 mol:mol) stoichiometry[1]. Blue, red and black symbols indicate N, Fe, and P addition respectively. Split circles indicate the addition of combined N+Fe (blue-red) or N + P (blue-black). Source data are provided in Supplementary Data 1.

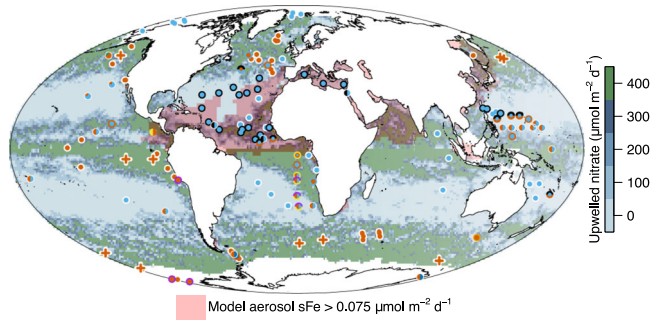

**Fig. 7 | Experimentally derived nutrient limitation patterns on a background of estimated nitrate upwelling.** Upwelled nitrate was calculated by multiplying the concentration of nitrate immediately below the mixed layer by wind-stress derived Ekman upwelling velocity. Regions of elevated soluble aerosol Fe deposition are highlighted. See Fig. 1 for symbol definitions. Source data are provided in Supplementary Data 1.

d). In contrast to expectations that might be based on stoichiometric consideration of phytoplankton demands, the magnitude of net growth following P addition actually decreased with increasing dissolved surface N:P (for log10(N:P), $R^2 = 0.34$; $p < 0.0001$, Fig. 6b; for temperature-normalized: log10(N:P), $R^2 = 0.35$; $p < 0.0001$). Despite this correlation, however, net growth responses to P addition always remained close to zero (mean = −0.02 $d^{-1}$; range = −0.48 to 0.55 $d^{-1}$),

consistent with the observation that P was not found to be the primary liming nutrient in any of the experiments (Fig. 1). Addition of N led to systematically higher net growth rates than P addition across the full range of surface dissolved N:P ratios encountered in the experimental dataset. Net growth rates following combined N and P addition were typically elevated above N alone at highest N:P (i.e., lowest P:N) ratios, consistent with a serially, and occasionally co-limiting, role for P observed in elevated N:P (i.e., P depleted) waters of the (sub)tropical North Atlantic and Mediterranean Sea (Fig. 1). However, the increase in net growth rates following combined N and P addition (maximum of 1.48 $d^{-1}$) were typically less enhanced than for combined N and Fe at other sites (reaching a maximum of 2.11 $d^{-1}$). There are two probable causes for these observations. Firstly, in contrast to N and P, N and Fe are often depleted to co-deficient levels (again based on assumed average phytoplankton nutrient requirements), whereas available surface water P was almost always in excess of N (Fig. 6b). A greater level of phytoplankton growth following sole N addition would therefore be expected to occur before P becomes serially limiting in comparison to the equivalent scenario for N and Fe. Secondly, highest growth responses to combined N and Fe addition for sites at or approaching N-Fe co-limitation have often been observed to be dominated by diatoms[11,13,26]. In contrast, responses to combined N + P addition in P depleted regions, such as the subtropical North Atlantic and Mediterranean, were typically dominated by the non-bloom forming picophytoplankton already dominating these systems[13,17,18]. These findings suggest that the type of system where N-P co-/serial

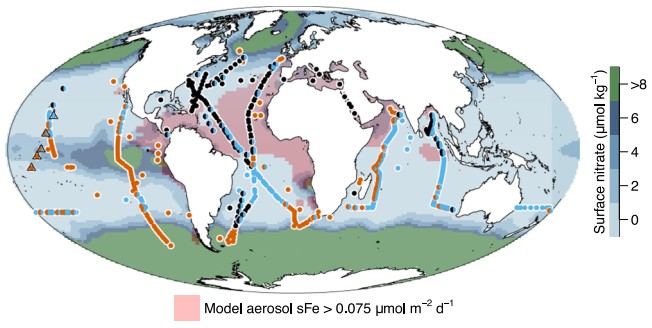

**Fig. 8 | Distribution of *Prochlorococcus* nutrient stress as suggested by molecular biomarkers.** Split symbols indicate co-stress by two nutrients. Data points are from the tropical Pacific study of ref. [57]. (highlighted with triangles) and ref. [61]. (all remaining data). Nutrient stressors in ref. [57]. were defined here as (i) the presence of P-II indicating N stress, (ii) the presence of idiA as indicating Fe stress, (iii) the presence of both as indicating N-Fe co-stress. For ref. [61], all data come from their principal component analysis of nutrient stress genes[61]. Background shading indicates climatological surface nitrate and aerosol Fe deposition (see Fig. 1).

limitation develops (i.e., highly oligotrophic with very low N supply rates and elevated Fe inputs favouring $N_2$ fixation and P drawdown)[18,29,53,54] in comparison to systems where N-Fe co-limitation develops (i.e., subtropical gyre boundaries)[11,13,26] therefore appears to be important in regulating the magnitude of growth responses to supply of limiting nutrient combinations in these different settings.

**Relating nutrient limitation and stress responses**

In addition to the limitation of overall phytoplankton biomass (or as a proxy, chlorophyll-a) accumulation registered in nutrient addition bioassay experiments, a diverse array of phytoplankton stress responses to nutrient scarcity have been observed in the ocean[55–61]. Whilst nutrient stress and limitation are potentially closely connected, these terms reflect distinct situations, with stress representing a physiological response to nutrient shortage that might or might not be growth-rate-limiting[1]. The recent publication of large-scale, internally consistent sets of nutrient stress biomarkers from proteomics[57] and genomics[61] for the abundant (sub-)tropical phytoplankton species *Prochlorococcus* provides a means to compare stress marker distributions with the synthesized bioassay dataset reported here at ocean basin scales. Both the proteomics (production of a nutrient-stress-specific biomarker protein) and genomics (in ref. [61], the retention or loss of a nutrient-stress-specific gene within a genome) results for *Prochlorococcus* show a broad-scale coherency with both the bioassay experimental results and calculated nutrient deficiency. Specifically, N stress (or at least selection for N stress related genes) is diagnosed in the stratified low latitude gyres, with Fe stress (or selection for Fe stress related genes) in upwelling regions (Fig. 8). Furthermore, evidence for N-Fe co-stress was registered by both genomic and proteomic approaches at the transition zones in between N and Fe limited provinces, again matching the results from the bioassay experimental dataset (Fig. 8).

A clear difference that does emerge between the stress and experimental limitation datasets related to P. Whilst the genomics analyses suggest that P is the nutrient leading to the strongest selective pressure on stress related gene retention in the (sub)tropical North Atlantic (Fig. 8), deficiency calculations (Fig. 5b, c) and the bioassay results both point towards primary N limitation, with P exerting a co-/serial limiting role (Fig. 1). Several hypotheses can be constructed to reconcile these observations. Firstly, stress related gene abundances from the genomic dataset are specific to *Prochlorococcus*, whilst bioassay experiments correspond to the whole phytoplankton community. However, a number of experiments in the (sub)tropical North Atlantic also measured *Prochlorococcus*-specific responses within

bioassay experiments and these demonstrated that *Prochlorococcus* showed similar responses to the bulk community (i.e., primary N limitation)[16–18,62,63]. Perhaps a more likely scenario is one where the substantial depletion of P in the (sub)tropical North Atlantic leads to the selective pressures for P stress related genes reflected in *Prochlorococcus* genome that, within a statistical analysis of overall genomic variability, dominate over the signal from the co-occurring N stress related genes observed throughout low N waters[61]. Ultimately, the related P stress responses might then be expected to act to reduce the impact of low P availability on phytoplankton growth in these regions[55,58], therefore buffering against P becoming the primary limiting nutrient. For example, the genomic data suggest substantial adaptations to low P conditions, including acquisition of P from the dissolved organic phosphorus (DOP) pool. Interestingly, DOP uptake might subsequently become regulated by availability of metal cofactors activating the responsible enzymes (e.g., Fe and Zn/Co in alkaline phosphatase[58,63,64]), leading to the potential for further co-limitation. Finally, from a stoichiometric perspective, it is worth noting that there appears to be considerably more flexibility in cellular P requirements than cellular N requirements[1,55].

**Outlook.** The analysis of synthesized nutrient enrichment bioassay responses presented aligns with previous findings demonstrating a broad-scale pattern of N limitation in the stratified, low-latitude ocean and Fe limitation in oceanic upwelling regions[1] but with N-Fe co-limitation at the boundaries between these systems. Co-/serial limiting roles for P, Mn, Co have been identified whereas for Zn and vitamin $B_{12}$, and silicic acid this is less clear due to the relatively few experiments where these nutrients have been supplied individually. Temporal variability in nutrient limitation, either on transient, seasonal or longer timescales, is largely unresolvable in the dataset (although see ref. [25]). Such variability is anticipated[65] and predicted by models[66–68]. Either establishment of time series of bioassay experimental observations and/or more concrete linkage of limitation to more easily measured assessments of nutrient stress[57,61] alongside their more widespread deployment are needed to address this issue.

We found that co- and serial limitation were common in the dataset (43% of experiments, which may be a lower estimate as not all tested for co-/serial limitation) across different systems (latitudes, temperature, trophic status). However, the magnitude of net growth responses following supply of limiting nutrients were elevated under warmer temperatures, likely due to higher maximum potential growth rates[42]. This temperature dependence of growth response to added nutrients, whilst expected, has potential implications for understanding ecosystem responses to natural perturbations of nutrient supply.

The coherent patterns in shifts from N to Fe limitation via regions of N-Fe co-limitation, the (co-)limiting role of Mn in parts of the Southern Ocean, and serial P limitation where P is depleted and Fe concentrations are elevated, are all consistent with predictions made by the seawater concentrations of these nutrients and their assumed-average requirements in phytoplankton[1,36,61]. Despite well-known variability in elemental requirements of different phytoplankton types and under different growth conditions[1], this coherency offers further support for well-established theoretical treatments of nutrient limitation[30,51] and hence a relatively simple numerical estimation of the nutrient limiting community-level phytoplankton growth in ecological ocean models, provided the nutrient is included in the model (not currently the case for Mn for instance, although see Ref. [37]). However, accurately modelling the realized phytoplankton growth rate under the specific nutrient limitation in question is recognized to be potentially substantially more complex, particularly at sub-community levels[8,69].

The database reported here generally provides strong support for coherency between new biomarker methods for detecting

nutrient stress and experimental determinations of nutrient limitation[57,61]. Such approaches offer a potential way to rapidly increase spatial-temporal resolution of nutrient limitation. Important differences, exemplified here by evidence for strong P selective pressure within systems experimentally determined to be N limited, suggest that work remains to be done to more completely link these approaches. The potential rapid expansion in deployment of such biomarker approaches, aided by their rapid throughput and via new programmes such as Biogeoscapes (www.biogeoscapes.org), underscores the value in reconciling such differences; for example, via coordinated studies deploying multiple limitation/stress assessment approaches and carrying out more detailed investigation of the responses of stress and biomass of individual phytoplankton types within bioassay experiments[31,59]. Such work may enable resolution of more subtle forms of biochemically-dependent and biochemical substitution co-limitation that may not be observable via biomass changes over relatively short incubation timescales[8,70]. Continued discovery and validation of appropriate stress biomarkers for more components of the phytoplankton community and for nutrients beyond N, P, and Fe are also required to link to results of bioassay experiments that have found limitation by these elements[31,32,38,40,71–74]. Ultimately a common weakness of both approaches is that neither omics nor bioassay approaches strictly reflect nutrient limitation of phytoplankton specific growth rate, the resolution of which for individual components of the phytoplankton community potentially remains an ultimate goal[6,69].

## Methods

### Bioassay experiment dataset

Bioassay experimental data, including chlorophyll-a concentrations, seawater nutrient concentrations, added nutrients and their concentrations, seawater temperatures, latitude, longitude, experimental duration, and limiting nutrient(s) were obtained from individual publications (Supplementary Data 1). Where not presented within the manuscript itself, these data were either: obtained directly from the authors of the relevant study; or a web-based data extractor tool was used to obtain data from publication figures (https://automeris.io/WebPlotDigitizer); or these data were not included in the dataset. New publications since the earlier compilation of ref. 1. were found using Google Scholar searches. Only studies where experimental treatments were conducted with a minimum of triplicate replication were included, aside from a limited number of single-replicate mesoscale in situ iron enrichment experiments that were also included for comparison. Qualitative nutrient limitation regimes (including primary, co-, and serial limitations) for each experiment were designated on the basis of assessments originally made in the individual study publications, which was almost always via the authors of these studies undertaking a statistical test (e.g., t-test or ANOVA followed by various post-hoc tests) to establish whether there were significant differences in mean chlorophyll-a concentrations between nutrient amended bottles and non-amended control bottles. We note the dependence of these results on (i) the chosen post-hoc test for assessing statistical differences between mean chlorophyll-a responses to treatments, and (ii) the related number of nutrient treatments included in the experiment, the combination of which will determine the relative likelihood of Type I or Type II errors[75]. A total of 159 assessments of nutrient limitation(s) at individual locations were included from this analysis.

### Net growth rate calculations

Chlorophyll-a concentrations measured within different nutrient treatments were next obtained from individual studies to independently quantify net growth responses to the different combinations of nutrient supply across the dataset as a whole. Net chlorophyll-a based

growth rate was chosen as the response metric, in order to account for differences across studies in (i) initial chlorophyll-a biomass; (ii) experimental durations[14,76]:

$$\mu_{Net}^{Chl} = \frac{\ln\left(\frac{Chl_T}{Chl_I}\right)}{t}, \quad (1)$$

Where $\mu_{Net}^{Chl}$ is the chlorophyll-a based net growth rate, $Chl_T$ is the chlorophyll-a concertation in the nutrient amended seawater or controls following incubation of length t (days), and $Chl_I$ is the initial chlorophyll-a concentration at the experimental start point ($t = 0$ days). To account for the expected increase in maximum potential growth rates ($\mu_{Max}^{Chl}$) with temperature, $\mu_{Max}^{Chl}$ was estimated using the equation from ref. 42

$$\mu_{Max}^{Chl} = 0.59 e^{0.0633T}, \quad (2)$$

Where T is the seawater temperature of each bioassay experiment. Subsequently $\mu_{Net}^{Chl}$ was normalized to the respective value of $\mu_{Max}^{Chl}$ to calculate a temperature-independent relative chlorophyll-a based net growth rate. A total of 765 absolute and 680 relative growth rate responses to nutrient limitation across the combined set of experiments and treatments were calculated in this manner (the lower number in the latter due to seawater temperatures not being available for some studies).

### Seawater nutrient deficiency calculations

The GEOTRACES IDP 2017 (version 2) nutrient dataset was downloaded from the BODC (https://www.bodc.ac.uk/geotraces/data/idp2017/; ref. 48). Sequential nutrient deficiency was calculated from surface seawater nutrient concentrations (samples < 10 m depth) by (1) normalizing seawater dissolved elemental concentrations to their respective assumed-average phytoplankton requirement from ref. 1, (2) arranging resulting values in ascending order, with the lowest value defining the most deficient nutrient. The employed stoichiometry was 16 N: 1 P: $7.5 \times 10^{-3}$ Fe: $2.8 \times 10^{-3}$ Mn: $8.0 \times 10^{-4}$ Zn :$1.9 \times 10^{-4}$ Co (ref. 1). Seawater concentrations of vitamin $B_{12}$ and silicic acid were not included in calculations as they (i) were generally not reported, and/or (ii) are required by only a few phytoplankton groups. For calculating deficiency predictions from the bioassay dataset, a criterion of a minimum of 2 measured nutrient concentrations was set.

### Nutrient stress biomarker datasets

For comparison to the experimental dataset presented here, the *Prochlorococcus* nutrient stress biomarker datasets of refs. 57,61 were obtained directly from the supplemental materials of these respective papers. Nutrient stressors in ref. 57. were defined here as (i) the presence of P-II indicating N stress, (ii) the presence of idiA as indicating Fe stress, (iii) the presence of both as indicating N-Fe co-stress. For the ref. 61, all data come from their principal component analysis of nutrient stress genes.

### Software and statistics

All data analysis and plotting was conducted with R software (R version 4.1.0 using packages 'base', 'stats', 'maps', 'mapdata', 'raster' and 'fields'). The kernel density estimates shown in Fig. 3b, d were calculated with the 'density' function from the 'stats' package.

### Reporting summary

Further information on research design is available in the Nature Portfolio Reporting Summary linked to this article.

## Data availability

The bioassay experiment dataset generated in this study (Supplementary Data 1) has been deposited in the Zenodo database under

accession code: https://doi.org/10.5281/zenodo.7937742. The GEO-TRACES Intermediate Data Product is available from the British Oceanographic Data Centre (BODC): https://www.bodc.ac.uk/geotraces/data/idp2017/.

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

## Acknowledgements

We thank Georgina Vickery for assisting with initial literature review and data extraction. T.J.B. acknowledges co-funding by the European Union (ERC, Ocean Glow, 101041453). Views and opinions expressed are however those of the author(s) only and do not necessarily reflect those of the European Union or the European Research Council. Neither the European Union nor the granting authority can be held responsible for them.

## Author contributions

T.J.B. and C.M.M. designed the study, performed the data analysis, and wrote the manuscript.

## Funding

## Competing interests

The authors declare no competing interests.
