## [Peer review file · Nature Communications]

REVIEWER COMMENTS

Reviewer #1 (Remarks to the Author):

This manuscript collects and analyzes a new global dataset of phytoplankton nutrient addition experiments to explore the patterns of nutrient limitation of phytoplankton growth at the global scale. The compilation of these results will be valuable to the oceanographic community. I found it to be a very interesting and well written manuscript, and I'm generally supportive of publication after several issues discussed below are addressed.

1 - I think the case for manganese as the primary limiting nutrient in the Southern Ocean is overstated. The evidence is kind of thin at this point. At best it could be co- or serially limiting with Fe right? There have been numerous in situ iron addition experiments in the Southern Ocean, all of which showed a massive response (with no Mn addition).

2 - Some mention/discussion of the Fe fertilization experiments is warranted. I don't think they are mentioned at all in main text. I would suggest including these as sites of demonstrated Fe-limitation in figure 1 (perhaps with a different figure shape).

3- Figure 3 is a little problematic. There was likely a strong tendency to add more than nutrients, precisely in the regions suspected of being co-limited...

4- lines 188-191 conclude that temperature is the primary driver of the growth rate response magnitude across the experiments. I don't think this conclusion is warranted.

It seems tough to separate from the latitudinal and starting chlorophyll trends. Several additional potential factors (that I find more convincing, but none of which are demonstrated) are noted by the authors in the following paragraph, initial biomass/nutrient levels, stronger grazing pressure in oligotrophic regions, etc...

Secondly, the Eppley (1972) temperature function, as shown in Fig. 4b doesn't appear to be a good way to try to normalize for temperature effects, not even as an upper bound. Presumably, nutrient stress is being relieved by nutrient addition, but the growth rates are nowhere near the Eppley Curve. Sherman et al. (2016) suggested a Q10 of 1.5 was more appropriate for mixed phytoplankton communities.

Figure 4 - what are the colors of the symbols indicating?, need a key.

5- nutrient stoichiometry section, Figure 5.

Please list the mean phytoplankton stoichiometry being used in these calculations. Given the growing evidence that phytoplankton reduced cellular quotas of the growth-limiting nutrient, is it appropriate to use the mean stoichiometry? For example, maybe assuming a mean N/P of 20 for the gyres might be better? Should there be a different "mean" stoichiometry for gyres versus non-gyres?

Is the ambient nutrient stoichiometry based just on nitrate?

That is was iron always at low concentrations, so the ratio mainly reflects nitrate?

This brings up a separate issue. In the gyres nitrate and phosphate are often below the detection level of traditional measurement techniques, so the ratio of these means little. Similarly, bioavailable N may be dominated by urea and ammonium, not nitrate in these regions. Similarly, phosphate and DOP are accessed by the phytoplankton.

The authors note that the stoichiometry of ambient nutrient concentrations are not the same as the supply stoichiometries, but then largely treat them as if they are the same thing. Iron inputs will likely always be uncoupled from the N and P inputs, due to the atmospheric source.

Maybe just a full paragraph discussing the caveats are needed for this section.

5 - P limitation

Does the metric of chlorophyll increase in the incubation experiments favor N-limitation over P-limitation? Given that there is significant flexibility in the C/P ratios of phytoplankton, maybe with P addition they initially just pack on more P per cell, whereas N addition is more likely to directly increase chlorophyll/photosystem activity? Growth might be co-limited, but N addition might lead to a larger chlorophyll response?

6 - modeling efforts?

Reading this manuscript one would think there has never been any attempt to model patterns of nutrient limitation at the global scale. Global ocean biogeochemical models capture well the broad patterns of nitrogen versus iron limitation highlighted in Figure 5, and are increasingly incorporating the

observed, variable plankton stoichiometry (Buchanan et al., 2018; Wang et al., 2019; Seferian et al., 2020; Hamilton et al., 2020; Tagliabue et al., 2020).

7 - Last point.. It seems to be assumed that the nutrient limitation patterns are largely static. Particularly in the regions which appear N-Fe or Fe-N serially-limited, the primary limiting nutrient likely shifts over seasonal to interannual timescales with varying atmospheric Fe inputs. On longer timescales the aerosol inputs (shown in Fig. 8) can vary drastically. Some discussion of the dynamic aspects of nutrient limitation patterns would be a good addition.

Here, too model results may be helpful in showing how patterns can shift under different climate regimes and atmospheric forcings (i.e. Krishnamurthy et al., 2010; Tagliabue et al., 2014; Buchanan et al., 2018; Hamilton et al., 2020).

Refs

Buchanan, P. J., Matear, R. J., Chase, Z., Phipps, S. J., & Bindoff, N. L. (2018). Dynamic biological functioning important for simulating and stabilizing ocean biogeochemistry. *Global Biogeochemical Cycles*, 32, 565–593.

Hamilton, D. S., Moore, J. K., Arneeth, A., Bond, T. C., Carslaw, K. S., Hantson, S., et al., (2020), Impact of changes to the atmospheric soluble iron deposition flux on ocean biogeochemical cycles in the Anthropocene. *Global Biogeochemical Cycles*, 34, e2019GB006448.

Krishnamurthy, A., Moore, J.K., Mahowald, N., Luo, C., and C.S. Zender, (2010), Impacts of atmospheric nutrient inputs on marine biogeochemistry, *J. Geophys. Res.*, 115, G01006, doi:10.1029/2009JG001115.

Seferian et al., (2020), Tracking Improvement in Simulated Marine Biogeochemistry Between CMIP5 and CMIP6. *Current Climate Change Reports* 6: 95-119.

Tagliabue, A., O. Aumont, and L. Bopp (2014), The impact of different external sources of iron on the global carbon cycle, *Geophys. Res. Lett.*, 41, 920–926, doi:10.1002/2013GL059059.

Tagliabue, A., Barrier, N., Du Pontavice, H., Kwiatkowski, L., Aumont, O., Bopp, L., et al. (2020). An iron cycle cascade governs the response of equatorial Pacific ecosystems to climate change. *Global Change Biology*. <https://doi.org/10.1111/gcb.15316>

Wang, W.L., Moore, J.K., Martiny, A.C., F.W. Primeau (2019). Convergent estimates of marine nitrogen fixation. *Nature*, 566, 205-211, <https://doi.org/10.1038/s41586-019-0911-2>.

Reviewer #2 (Remarks to the Author):

This is a nice synthesis of limitation and colimitation of ocean primary productivity, particularly as an update to the 2013 Moore et al. paper. Overall the synthesis is comprehensive and there are some interesting new observations. The figures will likely be popular in introductions of talks. I quite enjoyed reading and reviewing this and found it to be of broad interest and quite readable.

I recommend minor revisions, in particular some datasets that may be missing in the compilation (if they meet the criteria for inclusion), and the role of enzymes and metal substitution is largely absent in the discussion despite it being a potentially useful explanation for some multiple topics. These comments are expanded on below in the specific comments section.

Specific comments:

Line 132: The manuscript cites 1,3, and 3 silicic acid, Zn and B12 experiments in the literature. I believe these are underestimated by some references, listed below. Also it would be worth mentioning that there may be more negative experiments that have not been published:

There are additional Zn and B12 experiments that can be added:

Jakuba et al., GBC 2012 (Zn 2nd limitation North Pacific)

Franck et al., MEPS 2003 (Zn Fe in Costa Rica Dome)

Kellogg et al 2022 Nat Comms new Zn biomarkers and observation in South Pacific Ocean

Cohen et al., 2021 Nature Micro Fe and N Eukaryotic biomarkers in Central Pacific. Also this study has the full metal micronutrient data (except B12) for the Metzyme 2013 proteomic data used in this study (Fig 8 protein data use, Fig 5b nutrient stoichiometry predictions), if the authors do not already have it. The data are also in the latest GEOTRACES IDP.

Bertrand et al., 2012 Frontiers B12 Fe, B12, Co, Zn (Co 2nd negative for others)

Prior Zn negative results Cochlan 2002 Coale 2003 (perhaps excluded due to lack of replication?)

Pointing out the great difficulty of conducting trace metal limiting experiments, particularly with regards to Zn would be useful.

In addition, the section reviewing biomarkers is brief but useful. Some discussion about how biomarkers may be useful in increasing global spatial and temporal coverage of nutrient stress would be useful. The mention of BioGeoSCAPES effort is useful.

The discussion of Fe and P as secondary limiting nutrients, and the incongruence with the recent genomic biomarker dataset (especially in the North Atlantic) is notably lacking in discussion of the role of metals in acquisition of DOP. (As the authors know) Fe and Zn/Co are used in two isoforms of alkaline phosphatase that enables phytoplankton to use the much more abundant dissolved organic phosphorus. The observation that genomic data shows more P stress in the Atlantic then, can be interpreted as a larger genomic capacity for low P and DOP acquisition (as documented in Chisholm lab papers on P stress by Martiny and Coleman), which hence does not necessarily translate into P limitation due to the ability to switch from orthophosphate to DOP. The utility of genomic data (gene presence/absence) to infer stress/limitation is potentially worth bringing up, although could be construed as a (potentially valid?) criticism of that recent approach. This also brings into focus the alternate mechanism of biochemical substitution of co-limitation that isn't mentioned in the paper either. If useful there Prochlorococcus and Synechococcus specific alkaline phosphatase profiles in the back or Saito et al., 2020 BG <https://bg.copernicus.org/articles/14/4637/2017/>. Also the authors may want to cite Browning et al., 2017 and other studies(?) for the potential ecological role of metal replacement of APases.

Line 180: Do we not use the term Liebig limitation anymore?

Line 187: how is this a "theoretical" growth rate maxima? Because of the temperature correction? It's based on real data, so not really theoretical. T-corrected?

Line 189: "temperature is the main driver of the trends of absolute growth rate response to nutrient additions" This sentence seems misleading, as if temperature is the limiting parameter, not the nutrient.

There's some text about non-nutrient limited regions (light temp etc). This is an interesting topic that perhaps needs a bit more sunlight (excuse the pun). In some studies it is assumed the system is light limited. But in my opinion, it seems hard to believe anything is ever light limited, having seen some pea soup green places that are still nutrient (and iron) limited. To what extent is our knowledge of what is limiting itself limited by technical capabilities for micronutrient work? Does sole non-nutrient limitation (not co-limitation, clearly light-nutrient co-limitation exists as shown by Maldonado et al.) really exist in nature? I am skeptical. This is probably too big a can of worms to bring up (going back to the original L&O 1993 special issue debating bottom vs top down controls), but it seems to be lurking in the back of the intellectual thinking on the topic of limitation still.

Line 197: There are ways to get growth rate changes in grazed picoplankton, in particular the cell cycle method (Vaulot et al., 1997?). The resultant increase in growth rate without a change in Chlorophyll likely still has implications for the C cycle due to increased DOM production and release. To what extent is the experimental technique of incubations biasing these results and interpretations? More of an open question than expecting the authors to respond in the manuscript.

Figure 4 caption: Point out symbols are same color system as other figure(s).

Line 225: Nutrient stoichiometry calculations. There's almost no B12 data, so I assume B12 is excluded from this exercise? State what nutrients?

Line 235: Stoichiometric flexibility is brought up but not element substitution. Seems like that is relevant here?

Line 250: Interesting about Co and Zn. Kellogg et al., 2020 observed NE Pacific diatoms are particularly efficient for Co substitution for Zn, consistent with seawater Co:Zn. Not sure if that's quite at the boundaries being referred to. <https://doi.org/10.1002/lno.11471>

Line 268: "net growth rate decreased" This sounds like it is implying negative (death) rates as worded. But I think it means net growth rates increased less across this trend? This paragraph is a bit hard to follow due to the introduced concepts. Also line 323 negative slope of net growth rates. Maybe clarify.

Line 321-323: This sentence is dense and hard to follow, especially as an intro first sentence. The paragraph is a bit dense too, maybe some efforts to relate to the big picture a bit more.

Line 339: worthwhile to cite Ruttenberg or other early N:P correlation studies that observed this and point out this was thinking early on?

Line 330: "P was not found to be the primary limiting nutrient in any of the experiments" Presumably everything in low P environments is expending considerable resources to DOP acquisition. Why doesn't think happen for DON? Maybe the enzymes aren't as efficient DON isn't as readily available. Probably both. Interesting to think about mechanistically. Also probably reasonably beyond the scope of the synthesis paper.

Line 359: Biomarkers also work on *Synechococcus* (Saito et al., 2015 <https://doi.org/10.1002/pmic.201400630>) and Dinoflagellates (Cohen et al., 2020), and Diatoms (Bender et al., 2018).

Line 389: "Perhaps a more likely scenario is one where the substantial depletion of P in the sub(tropical) N. Atlantic leads to the selective pressures for P stress related genes ... in a statistical sense that overwhelms the co-occurring N stress related genes...". "statistical sense that overwhelms" seems kind of vague and perhaps unusual phrase. More satisfying explanations could include some discussion of the biochemical level, there are more means to spare and obtain P than N, given the differences in stoichiometry and usage. DOP multiple isoforms of APase, sulfolipids, versus DON usage from a more complex polymer (proteins vs DNA) require more enzymes to degrade. And perhaps the real science mystery is why so many of the P genes are all deleted in the Pacific (and outside of the N. Atlantic). The N acquisition genes seem to suffer less from deletion. But also this points to the risks of using genome-based analysis versus RNA or protein. Maybe this is more detail than the authors want to include, so not necessary for revisions if so, but might want to rephrase the "statistical sense that overwhelms".

Line 446: Could also add that there is limited experimental coverage due to time needed for experiments, high level of training and equipment for trace metal incubation experimentation, and limited knowledge of biomarkers and need for further discovery and validation work.

In general, the topic of speciation (N, P, metals) is not included. Maybe a few sentences somewhere? Maybe everything is accessible eventually is a gross simplification that geochemists sometimes like to invoke, (not sure I agree with, but maybe some fraction). But access to other species with specialized enzymes and transporters does have a resource cost.

Pointing out the limited number of experiments in general, and in availability of data in coastal regions in particular would be useful.

The subject of biomarkers is interesting and could be expanded on. Notably, the continued discovery and validation of biomarkers is needed. B12 and Zn biomarkers were recently discovered (Bertrand et al., 2013, Kellogg et al., 2022) in addition to the ones discussed in the cited studies (notably PII which was used here was considered unvalidated and somewhat controversial when we published the 2013 study). We did a review of available biomarkers (Table 2) and explanation for why they work that might be useful to refer to (Walworth J. Proteome Res. 2021. <https://doi.org/10.1021/acs.jproteome.1c00517>).

Another topic that the authors bring up is the disconnect between stress and growth rate. This is a useful statement for sure, but one aspect that could be added is that biomarkers or perhaps better phrased as “omic metrics” could be developed for growth rate itself, which would then bridge this gap, for example ribosomal or carbonic anhydrase enzymes may correlate with growth rate, if they can be further validated and calibrated (may need to be on a species-by-species basis). There is some early effort in this in the C13 biomarker field (Popp et al., 1997), and perhaps more recent molecular work.

Signed,

Mak Saito

Reviewer #3 (Remarks to the Author):

In this manuscript, the authors investigate the occurrence of nutrient limitation and co-limitation on phytoplankton on a global scale.

In general, the idea to summarize the existing data and extend them with new data in a systematic analysis is a great idea and highly relevant. However, in the abstract, it was not clear what exactly was done and what the question and the main findings were.

The abstract and the whole manuscript start with the bioassay experiments and data. This part is described in the methods (comments to methods see below) and the motivation is clear. It would be better if a clear ‘aim’ or ‘research question’ would be formulated. For the other parts, the methods are less detailed and clear (nutrient dataset) or not existing (molecular biomarkers, here only a reference is stated). For the third part, the molecular biomarkers, the motivation is not clear and the connection

between the three parts is not well presented. It reads as if there is the main part (experimental bioassays) and then some parts were added later. As the whole manuscript is very long I would recommend removing the third part. The second part needs more explanation of the methods and the motivation/connection to the first part needs to be clearer.

Another major comment is that the presentation of the figures needs to be improved. The readability is only given if the pdf version is zoomed in and it's not intuitive. I would recommend reducing the number of figures and revising the figures for better readability.

Detailed comments:

Abstract:

Line 20: 'a greater number of nutrients' sounds very unspecific. What does it mean? What was the investigated range of nutrients?

Introduction:

Lines 46-49: how can it be assumed that the impact of grazing was less in all studies than the addition of nutrients? Later the authors state that grazing can't be excluded.

Lines 60-65: Not clear what this means. Which more recent experimental programmes? Any other types of co-limitation to be considered?

Lines 70-75: Terms: why secondary P limitation? What does this mean and why is this term used here?

Methods:

Lines 461-462: what does this mean?

Lines 471-472: same here, what does this mean, how was that done? The experimental duration was included but what if the experimental duration had an impact on the Chl a concentrations?

Lines 478-480: Not clear how this correction for temperature could be applied for all regions. What about light or salinity-dependent effects on Chl a concentrations?

Lines 492-495: Unclear how this was done. The aim was also unclear; also how the data structure looked like. More details need to be provided here.

Results and discussion:

Line 83: Please explain how the 'nutrient limitation provinces' are refined. How did you calculate the percentage effects?

Adding the number of studies/data would be helpful to understand the impact of this information.

Lines 154-165: Which data are without and with temperature correction and why is it relevant to show both? Is it necessary to show both for all figures?

Lines 189-191: some statistical tests would help to strengthen this statement

Lines 203-205: Impact of grazers, see comment above. This should be clear in order to use the data.

Lines 221-222: more explanation is needed.

Lines 238-240: more explanation is needed.

Lines 264-277: unclear why this is relevant

Outlook:

This is a summary of the manuscript and some outlook is added. It is too long and has too much repetition of own results.

Figures:

Figure 1: Very difficult to read. The legend should be better explained.

Figure 2: The number of studies would be useful information here. Especially for the multiple nutrient addition experiments.

Figure 3: b&d: What does density mean here, this is not explained.

Figure 5: Legend: colors corresponding to Fig. 1 is not sufficient.

Figure 8: Figure and data (incl. discussion), see above, not clear why it is relevant here.

Response to Reviewer comments

We thank the three Reviewers for their detailed comments. We have responded to these below in blue.

To refer between review responses, we have numbered reviewer comments (R1_1, for Reviewer 1, comment 1, and so on).

Reviewer #1 (Remarks to the Author):

This manuscript collects and analyzes a new global dataset of phytoplankton nutrient addition experiments to explore the patterns of nutrient limitation of phytoplankton growth at the global scale. The compilation of these results will be valuable to the oceanographic community. I found it to be a very interesting and well written manuscript, and I'm generally supportive of publication after several issues discussed below are addressed.

We thank the reviewer for their supportive comments and have addressed the issues raised below in the revised manuscript.

1 - I think the case for manganese as the primary limiting nutrient in the Southern Ocean is overstated. The evidence is kind of thin at this point. At best it could be co- or serially limiting with Fe right? There have been numerous in situ iron addition experiments in the Southern Ocean, all of which showed a massive response (with no Mn addition).

R1_1: We agree with the reviewer that whilst there have been many experiments demonstrating iron limitation, only two (from one study) show primary manganese limitation. We have now modified the statement in the abstract with the aim to better bring out this uncertainty (see below). The remainder of discussion in the manuscript regarding manganese is minimal, so we have restricted changes to noting the co-/serial limitation role of Mn.

Abstract:

'Manganese can be co-limiting with iron in parts of the Southern Ocean...'

2 - Some mention/discussion of the Fe fertilization experiments is warranted. I don't think they are mentioned at all in main text. I would suggest including these as sites of demonstrated Fe-limitation in figure 1 (perhaps with a different figure shape).

R1_2: We have now included a statement in the Results and discussion noting that observations of Fe limitation are supported by chlorophyll and/or primary production increases in mesoscale iron enrichment experiments:

"The dataset is consistent with earlier reports in demonstrating widespread N limitation in the subtropical gyres, where surface N concentrations are depleted due to strong stratification of near-surface waters, and Fe limitation in the upwelling regions away from strong aerosol Fe sources, where N concentrations are elevated and Fe is often at low levels (Fig. 1)¹. Primary Fe limitation in the latter was furthermore supported by the chlorophyll and/or primary production increases observed in ten, kilometre-scale in situ Fe enrichment experiments, which are also included in the dataset (Fig. 1)."

Highlighting the mesoscale Fe fertilization experiments with different symbols is a good idea - this has now been done for Figure 1.

3- Figure 3 is a little problematic. There was likely a strong tendency to add more than nutrients, precisely in the regions suspected of being co-limited...

R1_3: We agree that most studies carrying out multi-nutrient treatments have been conducted in lower latitude waters where multiple nutrients are often scarce. We note this in the manuscript, and this is also highlighted in Figure 2. However, the multi-nutrient experiments that have been conducted do cover a very large geographic range, spanning upwelling regions, subtropical gyres and into temperate waters (see Fig. 2c), therefore we consider that the findings remain robust for the low-mid latitudes. We note in the manuscript the need for more multi-nutrient enrichment experiments at higher latitudes, particularly for the Southern Ocean where most nutrient addition experiments have been for Fe only.

4- lines 188-191 conclude that temperature is the primary driver of the growth rate response magnitude across the experiments. I don't think this conclusion is warranted. It seems tough to separate from the latitudinal and starting chlorophyll trends. Several additional potential factors (that I find more convincing, but none of which are demonstrated) are noted by the authors in the following paragraph, initial biomass/nutrient levels, stronger grazing pressure in oligotrophic regions, etc... Secondly, the Eppley (1972) temperature function, as shown in Fig. 4b doesn't appear to be a good way to try to normalize for temperature effects, not even as an upper bound. Presumably, nutrient stress is being relieved by nutrient addition, but the growth rates are nowhere near the Eppley Curve. Sherman et al. (2016) suggested a Q10 of 1.5 was more appropriate for mixed phytoplankton communities.

Figure 4 - what are the colors of the symbols indicating?, need a key.

R1_4: We agree with the reviewer's point regarding the difficulty in separating temperature from ecosystem characteristics, such as chlorophyll-a concentration. This prompted us to undertake a fuller statistical analysis that is now presented in the Supplementary Information (Supplementary Text 1), which offered further confirmation of this. We have now made modifications to the revised manuscript to highlight the challenge that co-varying temperature and ecosystem characteristics introduces into separating drivers from correlative parameters. We also suggest that whilst there is a strong theoretical and empirical reason to expect a temperature dependence of maximum, nutrient-replete growth rates, we cannot rule out, and indeed expect other co-varying ecosystem factors to be important (unfortunately with the data to hand it is difficult to present the latter in a more quantitative/statistical way).

We also assessed the different temperature dependence estimated by Sherman et al. (2016) suggested by the reviewer. Using the Sherman et al. relationship produced estimated maximal growth rates that were considerably below many of the observations in the dataset. Regardless, whilst impacting absolute values of normalized growth rates, from a correlative standpoint using either of the temperature normalizations has no impact on our conclusions.

The colours in Figure 4 indicate the number of added nutrients, which indeed was not highlighted in a key. A key has now been added to the figure/caption.

5- nutrient stoichiometry section, Figure 5.

Please list the mean phytoplankton stoichiometry being used in these calculations. Given the growing evidence that phytoplankton reduced cellular quotas of the growth-limiting nutrient, is it appropriate to use the mean stoichiometry? For example, maybe assuming a mean N/P of 20 for the gyres might be better? Should there be a different "mean" stoichiometry for gyres versus non-gyres?

Is the ambient nutrient stoichiometry based just on nitrate?

That is was iron always at low concentrations, so the ratio mainly reflects nitrate?

This brings up a separate issue. In the gyres nitrate and phosphate are often below the detection level of traditional measurement techniques, so the ratio of these means little. Similarly, bioavailable N may be dominated by urea and ammonium, not nitrate in these regions. Similarly, phosphate and DOP are accessed by the phytoplankton.

The authors note that the stoichiometry of ambient nutrient concentrations are not the same as the supply stoichiometries, but then largely treat them as if they are the same thing. Iron inputs will likely always be uncoupled from the N and P inputs, due to the atmospheric source.

Maybe just a full paragraph discussing the caveats are needed for this section.

R1_5: We step through each of the reviewer's points in turn below.

Please list the mean phytoplankton stoichiometry being used in these calculations.

The nutrient stoichiometry used in Figure 5 is now included in both the Figure 5 caption and Methods section.

Given the growing evidence that phytoplankton reduced cellular quotas of the growth-limiting nutrient, is it appropriate to use the mean stoichiometry? For example, maybe assuming a mean N/P of 20 for the gyres might be better? Should there be a different "mean" stoichiometry for gyres versus non-gyres?

We agree that flexible elemental stoichiometry is indeed an issue with this approach (i.e., seawater nutrient deficiency calculations), both in terms of reduced quotas (i.e., under nutrient limiting conditions) or higher quotas (i.e., under conditions of luxury nutrient uptake). The latter will also be related to shifts in phytoplankton community structure (that is, different inherent requirements of taxa in addition to within-taxa plasticity). Both plasticity in requirements together with differences in stoichiometry between phytoplankton taxa is poorly resolved, particularly for micronutrients.

A key outcome of this analysis is to demonstrate that even with the use of fixed elemental quotas derived from an average from a range of taxa grown under different conditions (replete and limiting nutrients, temperatures, light levels etc.), the approach taken generally produces very accurate predictions of the true limiting nutrient as defined by the bioassay experimental result. This goes some way in validating the broad link between dissolved seawater nutrient stoichiometry and limitation and thus, for example, provides a powerful indication of the potential for linking nutrient cycling and limitation in a tractable simplified way within numerical models. Clearly further refinement (particularly in the case of predicting co-limitation) may be possible in the future, for example as datasets describing stoichiometric variability in natural microbial populations reach comparable scales to the dissolved nutrient and experimental datasets we analysed.

Is the ambient nutrient stoichiometry based just on nitrate?

Due to the limited availability of ammonium concentrations, nitrate was always used for seawater N concentration.

That is was iron always at low concentrations, so the ratio mainly reflects nitrate?

Dissolved iron concentrations indeed vary much less than nitrate, so this largely drives the pattern between predicted (and observed) iron versus nitrogen limitation.

This brings up a separate issue. In the gyres nitrate and phosphate are often below the detection level of traditional measurement techniques, so the ratio of these means little. Similarly, bioavailable N may be dominated by urea and ammonium, not nitrate in these regions. Similarly, phosphate and DOP are accessed by the phytoplankton.

We agree that the availability of nanomolar-level data for nitrate and phosphate would be important for deriving highly accurate ratios. However, we think the available data are still of value: for example, throughout much of the low latitude oceans phosphate concentrations are actually above the detection limit (e.g., >50 nM) and therefore the prediction of N being more deficient than P in these regions is robust (see e.g., Deutsch et al. 2007 amongst many other compilations).

We also agree that other nutrient pools (ammonium, DON, DOP) will be regulating the true dissolved bioavailable seawater nutrient stoichiometry of N and P, but lack of quantification of their bioavailable concentrations (which are again potentially variable for different taxa) makes it very difficult to include in calculations such as these. This issue of bioavailability of different dissolved fractions also extends to other nutrients (e.g., Co, Ni). We have now included a comment on this in the revised manuscript:

“...Also, at such low concentrations, calculated residual surface water nutrient ratios will further be very sensitive to natural variability and measurement error associated with their quantification⁵². Thirdly, as indicated above, analysis on the basis of measured dissolved N and Fe again neglects the potential importance of nutrient speciation. For example, some of the measured trace metals might have limited bioavailability, while dissolved organic forms of nitrogen and phosphate were not considered, but might be bioavailable.. ...”

We also refer back to the point above that a first order stoichiometric treatment can be reconciled with the experimental data we compile, but there is clearly the potential for further refinement when larger datasets become available.

The authors note that the stoichiometry of ambient nutrient concentrations are not the same as the supply stoichiometries, but then largely treat them as if they are the same thing. Iron inputs will likely always be uncoupled from the N and P inputs, due to the atmospheric source.

We noted in the manuscript that from a theoretical perspective we would expect supply stoichiometries to be a better predictor of nutrient limitation, but under an assumption of steady state we would expect the residual concentrations to also have predictive power (see discussion in Ward et al. 2013 and Browning et al. 2017 for example). The interesting point raised by the reviewer regarding a decoupling of N, P and Fe with regards to atmospheric input (in comparison to e.g., subsurface turbulent diffusive fluxes) will be part of this: specifically, rather than standing surface ocean concentrations, the best prediction would be expected to be made using estimates of nutrient supply rates from the sub-surface and from aerosols and subsequently their stoichiometries. However, the data available to do these

calculations are very limited (e.g., with regards to aerosol deposition fluxes and the data required to calculate vertical fluxes) alongside the complexities associated with different operational timescales (e.g., transient aerosol fluxes versus seasonal winter mixing). Having stated this, the analysis associated within and presented in Figure 7 considers how the limitation patterns relate to surface water supplies.

References

- Browning, T.J., Achterberg, E.P., Rapp, I., Engel, A., Bertrand, E.M., Tagliabue, A. and Moore, C.M., 2017. Nutrient co-limitation at the boundary of an oceanic gyre. *Nature*, 551(7679), pp.242-246.
- Deutsch, C., Sarmiento, J.L., Sigman, D.M., Gruber, N. and Dunne, J.P., 2007. Spatial coupling of nitrogen inputs and losses in the ocean. *Nature*, 445(7124), pp.163-167.
- Ward, B.A., Dutkiewicz, S., Moore, C.M. and Follows, M.J., 2013. Iron, phosphorus, and nitrogen supply ratios define the biogeography of nitrogen fixation. *Limnology and Oceanography*, 58(6), pp.2059-2075.

6 - P limitation

Does the metric of chlorophyll increase in the incubation experiments favor N-limitation over P-limitation? Given that there is significant flexibility in the C/P ratios of phytoplankton, maybe with P addition they initially just pack on more P per cell, whereas N addition is more likely to directly increase chlorophyll/photosystem activity? Growth might be co-limited, but N addition might lead to a larger chlorophyll response?

R1_6: We fully agree with the reviewer on the potential importance of this. As the reviewer suggests, nitrogen addition to nitrogen-limited phytoplankton might stimulate more chlorophyll-a synthesis than phosphate addition to phosphate limited (or more likely N-P co-limited) phytoplankton. Moore et al. (2008, L&O) examined this in detail within the NP co-limited North Atlantic. This thus generates a potential issue with using chlorophyll-a as a proxy for phytoplankton biomass, which is what almost every study in the dataset used (although, as an aside, it could be argued that under a N-P co-limitation scenario, there might be little benefit to phytoplankton synthesizing light harvesting pigment following N addition if this could not be used for growth, see again Moore et al. 2008). As we mention in the last paragraph, this is one of the reasons why it would ultimately be better to have a technique to measure the response of phytoplankton specific growth rates to nutrient enrichment. This being said, in the specific case raised by the reviewer, two points make us believe that the phytoplankton in regions of low phosphate ((sub-)tropical North Atlantic and Mediterranean Sea) are still primarily nitrogen limited: (1) Where measured, net primary production responds in the same way as chlorophyll (i.e., to N alone but not to P), and (2) flow cytometry cell counts of dominant *Prochlorococcus* and *Synechococcus* also often show increases in cell concentrations in response to N but not P supply, reflecting overall carbon biomass as well as chlorophyll pigment changes (see e.g. Moore et al. 2008 L&O again for examples of this). The latter is mentioned in the manuscript (lines 434-437). Ultimately, as we state in the manuscript, the evidence suggests that these systems are at or close to NP co-limitation, but are clearly not primarily P limited.

7 - modeling efforts?

Reading this manuscript one would think there has never been any attempt to model patterns of nutrient limitation at the global scale. Global ocean biogeochemical models capture well the broad patterns of nitrogen versus iron limitation highlighted in Figure 5, and are increasingly incorporating the observed, variable plankton stoichiometry (Buchanon et al., 2018; Wang et al., 2019; Seferian et al., 2020; Hamilton et al., 2020; Tagliabue et al., 2020).

R1_7: An important contribution of the study is that it could be used as a useful benchmark of the accuracy of model predictions of the limiting nutrient. However, although we acknowledge the general point, we actually only partially agree with the reviewer's statement '*Global ocean biogeochemical models capture well the broad patterns of nitrogen versus iron limitation highlighted in Figure 5*'; for example, the different Earth System Model simulations assessed in Laufkötter et al. (2015) often strongly disagreed on the distributions of limiting nutrients; for example, their Figure 7 shows how out of seven of the compared global biogeochemical models, using the South Atlantic as an example, 4 models simulated nitrogen limitation, 1 phosphate limitation, and 2 iron limitation. Similar differences are observed in other oceanographic regions too. The simulated limiting nutrient could be important for regulating the model response to forcing (e.g., climate change simulations). We now highlight this as an additional justification for our study in the opening paragraph of the introduction.

"...Knowledge of the identity of these nutrients, and how their external supply impacts phytoplankton abundance and activity, are crucial for understanding and predicting the marine ecosystem responses to altered nutrient supply to the surface ocean, which may be associated with past and ongoing environmental changes¹⁻⁵. Such knowledge is subsequently key for the Earth System as a whole and carries strong economic and humanitarian importance, as phytoplankton activity regulates global nutrient cycles, atmosphere-ocean carbon exchange and the amount of carbon fixed and energy made available to higher trophic levels⁵. Understanding of nutrient limitation patterns is also important for rigorous assessment of Earth System Models, which still often disagree on the identity of limiting nutrients, at least at regional scales, potentially contributing to uncertainties in phytoplankton responses to climate change^{4,5}."

We also include reference to other modelling studies connected to the probable dynamic nature of nutrient limitation provinces (see below R1_8).

8 - Last point.. It seems to be assumed that the nutrient limitation patterns are largely static. Particularly in the regions which appear N-Fe or Fe-N serially-limited, the primary limiting nutrient likely shifts over seasonal to interannual timescales with varying atmospheric Fe inputs. On longer timescales the aerosol inputs (shown in Fig. 8) can vary drastically. Some discussion of the dynamic aspects of nutrient limitation patterns would be a good addition. Here, too model results may be helpful in showing how patterns can shift under different climate regimes and atmospheric forcings (i.e. Krishnamurthy et al., 2010; Tagliabue et al., 2014; Buchanan et al., 2018; Hamilton et al. ,2020).

R1_8: This is also a very good point: because of the difficulty/time/cost in conducting such experiments, these observations are largely static, with no seasonal or longer timescale resolution. We have now included this point in the revised 'Outlook' section.

"Temporal variability in nutrient limitation, either on transient, seasonal or longer timescales, is largely unresolvable in the dataset (although see Ref. 25). Such variability is anticipated⁶⁵ and predicted by models^{66,67,68}. Either establishment of time series of bioassay experimental observations and/or more concrete linkage of limitation to more easily measured assessments of nutrient stress^{57,61} alongside their more widespread deployment are needed to address this issue."

Refs

Buchanan, P. J., Matear, R. J., Chase, Z., Phipps, S. J., & Bindoff, N. L. (2018). Dynamic biological functioning important for simulating and stabilizing ocean biogeochemistry. *Global Biogeochemical Cycles*, 32, 565–593.

Hamilton, D. S., Moore, J. K., Arneeth, A., Bond, T. C., Carslaw, K. S., Hantson, S., et al., (2020), Impact of changes to the atmospheric soluble iron deposition flux on ocean biogeochemical cycles in the Anthropocene. *Global Biogeochemical Cycles*, 34, e2019GB006448.

Krishnamurthy, A., Moore, J.K., Mahowald, N., Luo, C., and C.S. Zender, (2010), Impacts of atmospheric nutrient inputs on marine biogeochemistry, *J. Geophys. Res.*, 115, G01006, doi:10.1029/2009JG001115.

Seferian et al., (2020), Tracking Improvement in Simulated Marine Biogeochemistry Between CMIP5 and CMIP6. *Current Climate Change Reports* 6: 95-119.

Tagliabue, A., O. Aumont, and L. Bopp (2014), The impact of different external sources of iron on the global carbon cycle, *Geophys. Res. Lett.*, 41, 920–926, doi:10.1002/2013GL059059.

Tagliabue, A., Barrier, N., Du Pontavice, H., Kwiatkowski, L., Aumont, O., Bopp, L., et al. (2020). An iron cycle cascade governs the response of equatorial Pacific ecosystems to climate change. *Global Change Biology*. <https://doi.org/10.1111/gcb.15316>

Wang, W.L., Moore, J.K., Martiny, A.C., F.W. Primeau (2019). Convergent estimates of marine nitrogen fixation. *Nature*, 566, 205-211, <https://doi.org/10.1038/s41586-019-0911-2>.

Reviewer #2 (Remarks to the Author):

This is a nice synthesis of limitation and colimitation of ocean primary productivity, particularly as an update to the 2013 Moore et al. paper. Overall the synthesis is comprehensive and there are some interesting new observations. The figures will likely be popular in introductions of talks. I quite enjoyed reading and reviewing this and found it to be of broad interest and quite readable.

I recommend minor revisions, in particular some datasets that may be missing in the compilation (if they meet the criteria for inclusion), and the role of enzymes and metal substitution is largely absent in the discussion despite it being a potentially useful explanation for some multiple topics. These comments are expanded on below in the specific comments section.

We thank Mak Saito for their positive evaluation of our manuscript and useful comments. We respond in turn to these below.

Specific comments:

Line 132: The manuscript cites 1,3, and 3 silicic acid, Zn and B12 experiments in the literature. I believe these are underestimated by some references, listed below. Also it would be worth mentioning that there may be more negative experiments that have not been published:

There are additional Zn and B12 experiments that can be added:

Jakuba et al., GBC 2012 (Zn 2nd limitation North Pacific)

Franck et al., MEPS 2003 (Zn Fe in Costa Rica Dome)

Kellogg et al 2022 Nat Comms new Zn biomarkers and observation in South Pacific Ocean

Cohen et al., 2021 Nature Micro Fe and N Eukaryotic biomarkers in Central Pacific. Also this study has the full metal micronutrient data (except B12) for the Metzzyne 2013 proteomic data used in this study (Fig 8 protein data use, Fig 5b nutrient stoichiometry predictions), if the authors do not already have it. The data are also in the latest GEOTRACES IDP.

Bertrand et al., 2012 Frontiers B12 Fe, B12, Co, Zn (Co 2nd negative for others)

Prior Zn negative results Cochlan 2002 Coale 2003 (perhaps excluded due to lack of replication?)

R2_1: We thank the reviewer for providing the additional references; a number of these were indeed studies that were considered but, in the end, not included in the dataset due to (1) lack of replication (our threshold was triplicate replication or higher, except for the mesoscale Fe enrichment experiments that were in and out of patch only; Jakuba et al. 2012; Cochlan et al., 2002; Coale et al., 2003) or (2) no measurement of either chlorophyll-a or primary production (Franck et al., 2003 Costa Rica Upwelling Dome site). We have now referred to the very useful supporting studies of Cohen et al., (2021) and Kellogg et al. (2022) in the biomarker part of the 'Outlook' section.

Pointing out the great difficulty of conducting trace metal limiting experiments, particularly with regards to Zn would be useful.

R2_2: We have now noted this in the revised manuscript (lines 48–50).

In addition, the section reviewing biomarkers is brief but useful. Some discussion about how biomarkers may be useful in increasing global spatial and temporal coverage of nutrient stress would be useful. The mention of BioGeoSCAPES effort is useful.

R2_3: We have now noted how biomarkers may be useful in increasing global spatial and temporal coverage of nutrient stress in the 'Outlook' section:

Firstly, lines 458–463:

“Temporal variability in nutrient limitation, either on transient, seasonal or longer timescales, is largely unresolvable in the dataset (although see Ref. 25). Such variability is anticipated^{d65} and predicted by models^{66,67,68}. Either establishment of time series of bioassay experimental observations and/or more concrete linkage of limitation to more easily measured assessments of nutrient stress^{57,61} alongside their more widespread deployment are needed to address this issue.”

Secondly, lines 491–502:

“The potential rapid expansion in deployment of such biomarker approaches, aided by their rapid throughput and via new programmes such as Biogeoscapes (www.biogeoscapes.org), underscores the value in reconciling such differences; for example, via coordinated studies deploying multiple limitation/stress assessment approaches and carrying out more detailed investigation of the responses of stress and biomass of individual phytoplankton types within bioassay experiments^{31,59}. Such work may enable resolution of more subtle forms of biochemically-dependent and biochemical substitution co-limitation that may not be observable via biomass changes over relatively short incubation timescales^{8,70}. Continued discovery and validation of appropriate stress biomarkers for more components of the phytoplankton community and for nutrients beyond N, P, and Fe are also required to link to results of bioassay experiments that have found limitation by these elements^{31,32,38,40,71-74}.”

The discussion of Fe and P as secondary limiting nutrients, and the incongruence with the recent genomic biomarker dataset (especially in the North Atlantic) is notably lacking in discussion of the role of metals in acquisition of DOP. (As the authors know) Fe and Zn/Co are used in two isoforms of alkaline phosphatase that enables phytoplankton to use the much more abundant dissolved organic phosphorus. The observation that genomic data shows more P stress in the Atlantic then, can be interpreted as a larger genomic capacity for low P and DOP acquisition (as documented in Chisholm lab papers on P stress by Martiny and Coleman), which hence does not necessarily translate into P limitation due to the ability to switch from orthophosphate to DOP. The utility of genomic data (gene presence/absence) to infer stress/limitation is potentially worth bringing up, although could be construed as a (potentially valid?) criticism of that recent approach. This also brings into focus the alternate mechanism of biochemical substitution of co-limitation that isn't mentioned in the paper either. If useful there Prochlorococcus and Synechococcus specific alkaline phosphatase profiles in the back or Saito et al., 2020

BG <https://bg.copernicus.org/articles/14/4637/2017/>. Also the authors may want to cite Browning et al., 2017 and other studies(?) for the potential ecological role of metal replacement of APases.

R2_4: We have now made the following additions to include these aspects:

Firstly, within the stress response section:

“For example, the genomic data suggest substantial adaptations to low P conditions, including acquisition of P from the dissolved organic phosphorus (DOP) pool. Interestingly, DOP uptake might subsequently become regulated by availability of metal co-factors

activating the responsible enzymes (e.g., Fe and Zn/Co in alkaline phosphatase^{58,63,64}, leading to the potential for further co-limitation.”

Secondly, within the ‘Outlook’ section:

“The potential rapid expansion in deployment of such biomarker approaches, aided by their rapid throughput and via new programmes such as Biogeoscapes (www.biogeoscapes.org), underscores the value in reconciling such differences; for example, via coordinated studies deploying multiple limitation/stress assessment approaches and carrying out more detailed investigation of the responses of stress and biomass of individual phytoplankton types within bioassay experiments^{31,59}. Such work may enable resolution of more subtle forms of biochemically-dependent and biochemical substitution co-limitation that may not be observable via biomass changes over relatively short incubation timescales^{8,70}.”

Line 180: Do we not use the term Liebig limitation anymore?

R2_5: We avoided the use of this term (alongside Blackman) as we considered it well covered elsewhere (e.g., Saito et al. 2008; Moore et al. 2013)

Line 187: how is this a “theoretical” growth rate maxima? Because of the temperature correction? It’s based on real data, so not really theoretical. T-corrected?

R2_6: This was referring to the maximum growth rate predicted by the Eppley (1972) equation, which describes the upper envelope of phytoplankton growth rates measured under different temperatures (Equation 2 in Methods). We recognise the lack of clarity and rephrased this to:

“Correspondingly, when normalized to potential growth rate maxima predicted by ambient temperature...”

Line 189: “temperature is the main driver of the trends of absolute growth rate response to nutrient additions” This sentence seems misleading, as if temperature is the limiting parameter, not the nutrient.

R2_7: Now rephrased to:

“This suggested that, following addition of the limiting nutrient(s), temperature is potentially the main driver of the trends of absolute growth rate response (Fig. 4b)...”

There’s some text about non-nutrient limited regions (light temp etc). This is an interesting topic that perhaps needs a bit more sunlight (excuse the pun). In some studies it is assumed the system is light limited. But in my opinion, it seems hard to believe anything is ever light limited, having seen some pea soup green places that are still nutrient (and iron) limited. To what extent is our knowledge of what is limiting itself limited by technical capabilities for micronutrient work? Does sole non-nutrient limitation (not co-limitation, clearly light-nutrient co-limitation exists as shown by Maldonado et al.) really exist in nature? I am skeptical. This is probably too big a can of worms to bring up (going back to the original L&O 1993 special issue debating bottom vs top down controls), but it seems to be lurking in the back of the intellectual thinking on the topic of limitation still.

R2_8: We agree that the collective impact of nutrients and light still needs better resolution, also in the context of grazing pressure. Some studies have shown what we consider to be fairly convincing evidence for ‘no nutrient limitation’, in that they demonstrate (1) strong

increases in phytoplankton biomass in non-nutrient-amended control treatments over initial conditions and (2) no further positive response to any of the added nutrients expected to be (co-)limiting (for example, Experiment D350.2 in Ryan-Keogh et al., 2013; Experiment 1 in Browning et al., 2017; Experiment 3 in Browning et al., 2018; Experiment 10 in Browning et al., 2021). However, it should be noted that the assumption that there was no change in light, micronutrient contamination, or grazing in the control bottles, which is difficult to concretely rule out, needs to be factored into interpretation of such results.

References

- Browning, T.J., Achterberg, E.P., Rapp, I., Engel, A., Bertrand, E.M., Tagliabue, A. and Moore, C.M., 2017a. Nutrient co-limitation at the boundary of an oceanic gyre. *Nature* **551**, 242-246.
- Browning, T.J., Rapp, I., Schlosser, C., Gledhill, M., Achterberg, E.P., Bracher, A. and Le Moigne, F.A., 2018. Influence of iron, cobalt, and vitamin B12 supply on phytoplankton growth in the tropical East Pacific during the 2015 El Niño. *Geophysical Research Letters*, *45*(12), pp.6150-6159.
- Browning, T.J., Achterberg, E.P., Engel, A. and Mawji, E., 2021. Manganese co-limitation of phytoplankton growth and major nutrient drawdown in the Southern Ocean. *Nat. Commun.* **12**, 884.
- Ryan-Keogh, T.J., Macey, A.I., Nielsdóttir, M.C., Lucas, M.I., Steigenberger, S.S., Stinchcombe, M.C., Achterberg, E.P., Bibby, T.S. and Moore, C.M., 2013. Spatial and temporal development of phytoplankton iron stress in relation to bloom dynamics in the high-latitude North Atlantic Ocean. *Limnol. Oceanogr.* **58**, 533-545.

Line 197: There are ways to get growth rate changes in grazed picoplankton, in particular the cell cycle method (Vaulot et al., 1997?). The resultant increase in growth rate without a change in Chlorophyll likely still has implications for the C cycle due to increased DOM production and release. To what extent is the experimental technique of incubations biasing these results and interpretations? More of an open question than expecting the authors to respond in the manuscript.

R2_9: We agree this is an important point. We later comment on this in the 'Outlook' section as a weakness of using changes in phytoplankton biomass (or a proxy thereof) as a response variable rather than growth rate (lines 502–505 of the revised manuscript).

Figure 4 caption: Point out symbols are same color system as other figure(s).

R2_10: Thanks for pointing this out – a legend is now also added to this figure.

Line 225: Nutrient stoichiometry calculations. There's almost no B12 data, so I assume B12 is excluded from this exercise? State what nutrients?

R2_11: This is correct – as a result of almost no dissolved vitamin B₁₂ concentration data, this nutrient was excluded. This has now been indicated, and an additional note of which nutrients were used for the calculations and the utilized phytoplankton elemental stoichiometry has been included, in both the Figure 5 legend and the Methods section.

Line 235: Stoichiometric flexibility is brought up but not element substitution. Seems like that is relevant here?

R2_12: We intended 'flexibility' to encompass all mechanisms leading to changes in elemental stoichiometry – we have now clarified this:

“Prediction of nutrient co-limitation (i.e., ‘iii’ above) from dissolved concentrations alone is similarly challenging. Once again, residual nutrient standing stocks following biological removal should provide an indicator of the transitions between regions of single nutrient limitation and co-limitation¹¹. However, significant biological stoichiometric flexibility (including elemental substitutions⁸) potentially needs to be taken into account. However, it might be possible to set ranges in nutrient deficiency around the average stoichiometry where both nutrients are equally limiting¹¹.”

Line 250: Interesting about Co and Zn. Kellogg et al., 2020 observed NE Pacific diatoms are particularly efficient for Co substitution for Zn, consistent with seawater Co:Zn. Not sure if that’s quite at the boundaries being referred to. <https://doi.org/10.1002/lno.11471>

R2_13: Kellogg et al. (2020) is now cited in the biomarker part of the ‘Outlook’ section.

Line 268: “net growth rate decreased” This sounds like it is implying negative (death) rates as worded. But I think it means net growth rates increased less across this trend? This paragraph is a bit hard to follow due to the introduced concepts. Also line 323 negative slope of net growth rates. Maybe clarify.

R2_14: We agree on re-reading that this phrasing gave this false interpretation. Now rephrased to:

“At progressively increasing dissolved N:Fe, the magnitude of net growth following N addition decreased (for $\log_{10}(N:Fe)$, $R^2=0.20$; $p=0.00012$), whilst the magnitude of net growth following Fe addition increased (for $\log_{10}(N:Fe)$, $R^2=0.24$; $p<0.0001$).”

And:

“In contrast to expectations that might be based on stoichiometric consideration of phytoplankton demands, the magnitude of net growth following P addition actually decreased with increasing dissolved surface N:P (for $\log_{10}(N:P)$, $R^2=0.34$; $p<0.0001$, Fig. 6b; for temperature-normalized: $\log_{10}(N:P)$, $R^2=0.35$; $p<0.0001$).”

Line 321-323: This sentence is dense and hard to follow, especially as an intro first sentence. The paragraph is a bit dense too, maybe some efforts to relate to the big picture a bit more.

R2_15: We have now split up the first sentence of the paragraph. This now reads:

“In theory, the greatest enhancement of net growth rates following combined N+Fe additions relative to individual N and/or Fe additions should be observed at intermediate dissolved N:Fe ratios^{10,11,51}. This intermediate dissolved N:Fe ratio should approximately correspond to the intersection point of the net growth rate–dissolved N:Fe ratio slopes for individual N and Fe additions and the value of typical phytoplankton N:Fe requirements, which appeared to match well with each other in our analysis (intersection of red, green and dashed lines in Fig. 6a,c)^{10,11,51}.”

Line 339: worthwhile to cite Ruttenberg or other early N:P correlation studies that observed this and point out this was thinking early on?

R2_16: We thank the reviewer for the suggestion, but cannot specifically see which part in line 339 this should refer to (we were also unsure which Ruttenberg paper the reviewer was referring to). In this instance we have therefore not added any additional reference.

Line 330: “P was not found to be the primary limiting nutrient in any of the experiments” Presumably everything in low P environments is expending considerable resources to DOP acquisition. Why doesn’t think happen for DON? Maybe the enzymes aren’t as efficient DON isn’t as readily available. Probably both. Interesting to think about mechanistically. Also probably reasonably beyond the scope of the synthesis paper.

R2_17: We agree with the reviewers comment, both in terms of the additional N required for protein synthesis (e.g., for AP enzymes) and metals (e.g., Fe and Zn/Co for AP) under conditions of low DIP. But we think the statement is consistent with our definition of limitation in the manuscript (i.e., no experiment observed an increase in chlorophyll-a biomass or PP in response to P-only addition over non-amended controls). As the reviewer indicates, we prefer not to speculate further on this within the current manuscript.

Line 359: Biomarkers also work on *Synechococcus* (Saito et al., 2015 <https://doi.org/10.1002/pmic.201400630>) and Dinoflagellates (Cohen et al., 2020), and Diatoms (Bender et al., 2018).

R2_18: Citations to these studies have now been included in the biomarker part of the ‘Outlook’ section.

Line 389: “Perhaps a more likely scenario is one where the substantial depletion of P in the sub(tropical) N. Atlantic leads to the selective pressures for P stress related genes ... in a statistical sense that overwhelms the co-occurring N stress related genes...”. “statistical sense that overwhelms” seems kind of vague and perhaps unusual phrase. More satisfying explanations could include some discussion of the biochemical level, there are more means to spare and obtain P than N, given the differences in stoichiometry and usage. DOP multiple isoforms of APase, sulfolipids, versus DON usage from a more complex polymer (proteins vs DNA) require more enzymes to degrade. And perhaps the real science mystery is why so many of the P genes are all deleted in the Pacific (and outside of the N. Atlantic). The N acquisition genes seem to suffer less from deletion. But also this points to the risks of using genome-based analysis versus RNA or protein. Maybe this is more detail than the authors want to include, so not necessary for revisions if so, but might want to rephrase the “statistical sense that overwhelms”.

R2_19: We used the phrase ‘in the statistical sense’ as Ustick et al. (2021) draw their conclusions as to whether a given sample/location were considered N, P, or Fe limited depending on the principal component analysis conducted with the genomics dataset, i.e., it was statistically based. However, we have now added to this section:

*“Perhaps a more likely scenario is one where the substantial depletion of P in the (sub)tropical North Atlantic leads to the selective pressures for P stress related genes reflected in *Prochlorococcus* genome that, within a statistical analysis of overall genomic variability, dominate over the signal from the co-occurring N stress related genes observed throughout low N waters⁶¹. Ultimately, the related P stress responses might then be expected to act to reduce the impact of low P availability on phytoplankton growth in these regions^{55,58}, therefore buffering against P becoming the primary limiting nutrient. For example, the genomic data suggest substantial adaptations to low P conditions, including acquisition of P from the dissolved organic phosphorus (DOP) pool. Interestingly, DOP*

uptake might subsequently become regulated by availability of metal co-factors activating the responsible enzymes (e.g., Fe and Zn/Co in alkaline phosphatase^{58,63,64}, leading to the potential for further co-limitation. Finally, from a stoichiometric perspective, it is worth noting that there appears to be considerably more flexibility in cellular P requirements than cellular N requirements^{1,55}.”

Line 446: Could also add that there is limited experimental coverage due to time needed for experiments, high level of training and equipment for trace metal incubation experimentation, and limited knowledge of biomarkers and need for further discovery and validation work.

R2_20: These parts have now been added earlier in the revised ‘Outlook’ section (lines 458–463 and 488–502).

In general, the topic of speciation (N, P, metals) is not included. Maybe a few sentences somewhere? Maybe everything is accessible eventually is a gross simplification that geochemists sometimes like to invoke, (not sure I agree with, but maybe some fraction). But access to other species with specialized enzymes and transporters does have a resource cost.

R2_21: We have now briefly referred to this alongside the request for noting DON/P and ammonium by Reviewer 1 (R1_5) in lines 351–354 of the revised manuscript:

“Thirdly, as indicated above, analysis on the basis of measured dissolved N and Fe again neglects the potential importance of nutrient speciation. For example, some of the measured trace metals might have limited bioavailability, while dissolved organic forms of nitrogen and phosphate were not considered, but might be bioavailable.”

Pointing out the limited number of experiments in general, and in availability of data in coastal regions in particular would be useful.

R2_22: The need for more experiments (and or further linking and expansion via biomarkers) is now referred to in the revised ‘Outlook’ section (lines 456–460).

The subject of biomarkers is interesting and could be expanded on. Notably, the continued discovery and validation of biomarkers is needed. B12 and Zn biomarkers were recently discovered (Bertrand et al., 2013, Kellogg et al., 2022) in addition to the ones discussed in the cited studies (notably PII which was used here was considered unvalidated and somewhat controversial when we published the 2013 study). We did a review of available biomarkers (Table 2) and explanation for why they work that might be useful to refer to (Walworth J. Proteome Res. 2021. <https://doi.org/10.1021/acs.jproteome.1c00517>).

R2_23: We have now noted this in the ‘Outlook’ section, including the citation to the Walworth study (useful synthesis table):

“The potential rapid expansion in deployment of such biomarker approaches, aided by their rapid throughput and via new programmes such as Biogeoscapes (www.biogeoscapes.org), underscores the value in reconciling such differences; for example, via coordinated studies deploying multiple limitation/stress assessment approaches and carrying out more detailed investigation of the responses of stress and biomass of individual phytoplankton types within bioassay experiments^{31,59}. Such work may enable resolution of more subtle forms of biochemically-dependent and biochemical substitution co-limitation that may not be observable via biomass changes over relatively short incubation timescales^{8,70}. Continued

discovery and validation of appropriate stress biomarkers for more components of the phytoplankton community and for nutrients beyond N, P, and Fe are also required to link to results of bioassay experiments that have found limitation by these elements^{31,32,38,40,71-74}."

Another topic that the authors bring up is the disconnect between stress and growth rate. This is a useful statement for sure, but one aspect that could be added is that biomarkers or perhaps better phrased as "omic metrics" could be developed for growth rate itself, which would then bridge this gap, for example ribosomal or carbonic anhydrase enzymes may correlate with growth rate, if they can be further validated and calibrated (may need to be on a species-by-species basis). There is some early effort in this in the C13 biomarker field (Popp et al., 1997), and perhaps more recent molecular work.

R2_24: We very much agree with this; such 'omic metrics' for growth rate would be of invaluable utility if reliable (acknowledging that this might be very challenging given the potential need for species-, or even ecotype-, specific temperature/light/nutrient calibrations!). As a pointer towards this we cited the recent study of McCain et al. (2021) that attempted this with some success (via a model).

Signed,
Mak Saito

Reviewer #3 (Remarks to the Author):

In this manuscript, the authors investigate the occurrence of nutrient limitation and co-limitation on phytoplankton on a global scale.

In general, the idea to summarize the existing data and extend them with new data in a systematic analysis is a great idea and highly relevant. However, in the abstract, it was not clear what exactly was done and what the question and the main findings were.

The abstract and the whole manuscript start with the bioassay experiments and data. This part is described in the methods (comments to methods see below) and the motivation is clear. It would be better if a clear 'aim' or 'research question' would be formulated. For the other parts, the methods are less detailed and clear (nutrient dataset) or not existing (molecular biomarkers, here only a reference is stated). For the third part, the molecular biomarkers, the motivation is not clear and the connection between the three parts is not well presented. It reads as if there is the main part (experimental bioassays) and then some parts were added later. As the whole manuscript is very long I would recommend removing the third part. The second part needs more explanation of the methods and the motivation/connection to the first part needs to be clearer.

R3_1: We thank the reviewer for their evaluation of our manuscript and respond to each comment below.

We consider that what was done in the study (synthesis of available experimental data) and what we found is clear in the abstract. We do agree that a concrete set of aims was not listed here (in the abstract), but at the end of the introduction due to space constraints. We also agree with the reviewer that the latter should include a link to the nutrient stoichiometry and nutrient stress biomarker sections of the manuscript. These have now been added. Revised section:

“Our primary goals were to: (i) add to the spatial extent and resolution of experimentally-determined nutrient limitation in the global ocean; (ii) place recent findings of both co-limitation and limitation by nutrients other than N, P, and Fe into a global context; (iii) quantitatively evaluate differential phytoplankton growth responses to nutrient supply and dissect potential driving factors; (iv) evaluate qualitative and quantitative responses to nutrient treatments in the context of ambient seawater nutrient concentrations, and (v) compare meta-analysis of experimental data to recent molecular biomarker datasets of nutrient stress.”

Our justification for comparing to nutrient concentrations, and calculated deficiencies, is that this was expected a priori to be a major factor regulating the phytoplankton response to experimental nutrient supply. Indeed, this was the case, as the nutrient deficiency calculations made excellent predictions of which nutrient was limiting. We think this is highly relevant, as it goes some way to validate theoretical links between dissolved seawater nutrient stoichiometry and nutrient limitation (noting caveats with more subtle co-limitations discussed in the responses to Reviewer 2), as well as providing support for more easily measured nutrient concentrations being used to make much more widespread predictions of nutrient limitation (i.e., to extend observations geographically and temporally). We think the link between the bioassay experiment results section and the nutrient stoichiometry section is quite clear (e.g., first line of the latter section is *‘Clear linkages were found between the nutrient found to be limiting experimentally and the nutrient predicted to be most deficient (Fig. 5)^{1,36}*, which is then followed by a description of the nutrient deficiency calculation method).

The justification for comparing our results to molecular biomarkers of nutrient stress was to investigate (in)consistencies between the two assessments. It is perhaps worth reiterating here that we define stress as a physiological response to nutrient shortage (including presence of a stress protein or retention of a gene in a genome) whereas we define limitation as the occurrence of a positive biomass increase in response to nutrient supply. We might expect there to be both similarities and differences between nutrients found to induce stress responses and those that are limiting, and these could provide important information about the links between microbial physiology and limiting nutrients. For this, we decided to use recent datasets that focussed on *Prochlorococcus*, as these datasets focussed specifically on nutrient stress and were geographically widespread. As we point out in the manuscript: there is both excellent match-up with regards to N and Fe (co-) limitation/stress, but a discrepancy with regards to phosphate. Whilst not the major contribution of the manuscript, we feel that this relatively short section is of value in making this comparison. As for the nutrient stoichiometry, this analysis also set up ways forward for expanding observations geographically and temporally, discussed in the 'Outlook' section, for example, in order to observe the impacts of climate change on nutrient limitation.

We have now moderately expanded the methods sections for both (i) nutrient deficiency calculations, (ii) explanation of the molecular biomarker datasets used to compare to the experimental dataset. These changes are described in more detail below in response to specific comments (R3_10 and R3_23).

Another major comment is that the presentation of the figures needs to be improved. The readability is only given if the pdf version is zoomed in and it's not intuitive. I would recommend reducing the number of figures and revising the figures for better readability.

R3_2: The figures have now all been revised for clarity. We have retained all figures as we consider each of them to provide important support of points raised in the manuscript.

Detailed comments:

Abstract:

Line 20: 'a greater number of nutrients' sounds very unspecific. What does it mean? What was the investigated range of nutrients?

R3_3: This referred to the number of added nutrients (e.g., 1, 2, 3, or 4 added nutrients). Potentially there was a confusion with greater amount of added nutrient (i.e., a higher added concentration). We have now tried to make this clearer by rephrasing to 'different nutrients':

"Overall, a metaanalysis of experimental responses showed that phytoplankton net growth can be significantly enhanced through increasing the number of different nutrients supplied..."

Introduction:

Lines 46-49: how can it be assumed that the impact of grazing was less in all studies than the addition of nutrients? Later the authors state that grazing can't be excluded.

R3_4: We agree that the potential for differential grazing between nutrient treatments cannot be unequivocally excluded (i.e., that the grazer response to the initial ecophysiological changes induced within the experiments lead to knock on effects on grazing pressure); this is an assumption made in all nutrient-addition bioassay experiments assessing before and after incubation biomass changes. That being said, the typical high level of coherency in responses to nutrient supply (both in terms of replicates within a given treatment and

between different nutrient combinations) suggests it is, in general, a fairly robust assumption.

Lines 60-65: Not clear what this means. Which more recent experimental programmes? Any other types of co-limitation to be considered?

R3_5: References have now been added accordingly to cite 'existing experimental compilations' and 'more recent experimental programmes' (we agree original phrasing was poor, this has now been changed to 'more recent experiments'). Essentially, we are trying to say that both a past experimental dataset (Moore et al., 2013) and new published studies suggest co- and serial nutrient limitation can be widespread, and our aim here is to put this into a more synthesized perspective.

The resolution of other co-limitations (e.g., biochemical substitution, or biochemically-dependent co-limitation; Saito et al., 2008) are limited to only a few experiments that relied on data other than chlorophyll-a biomass or bulk net primary production rates to infer the type of co-limitation (e.g., Fe/P limitation of N₂ fixation; Mills et al., 2004; Wen et al., 2022; Fe or Zn limitation of alkaline phosphatase activity; Mahaffey et al., 2014; Browning et al., 2017), as chlorophyll-a biomass changes did not accompany these.

References

- Browning, T.J., Achterberg, E.P., Yong, J.C., Rapp, I., Utermann, C., Engel, A. and Moore, C.M., 2017. Iron limitation of microbial phosphorus acquisition in the tropical North Atlantic. *Nature communications*, 8(1), p.15465.
- Mahaffey, C., Reynolds, S., Davis, C.E. and Lohan, M.C., 2014. Alkaline phosphatase activity in the subtropical ocean: insights from nutrient, dust and trace metal addition experiments. *Frontiers in Marine Science*, 1, p.73.
- Mills, M.M., Ridame, C., Davey, M., La Roche, J. and Geider, R.J., 2004. Iron and phosphorus co-limit nitrogen fixation in the eastern tropical North Atlantic. *Nature* **429**, 292-294.
- Saito, M.A., Goepfert, T.J. and Ritt, J.T., 2008. Some thoughts on the concept of colimitation: three definitions and the importance of bioavailability. *Limnol. Oceanogr.* **53**, 276-290.
- Wen, Z., Browning, T.J., Cai, Y., Dai, R., Zhang, R., Du, C., Jiang, R., Lin, W., Liu, X., Cao, Z. and Hong, H., 2022. Nutrient regulation of biological nitrogen fixation across the tropical western North Pacific. *Science Advances*, 8(5), p.eabl7564.

Lines 70-75: Terms: why secondary P limitation? What does this mean and why is this term used here?

R3_6: We apologize that this was not clear. We have now switched this phrasing to 'serial P limitation', with serial limitation being defined in the previous paragraph.

['Secondary' limitation refers to a specific case of serial limitation, whereby a subsequent biomass enhancement is observed following the combined addition of a second nutrient with the primary limiting nutrient.]

Methods:

Lines 461-462: what does this mean?

R3_7: We have now tried to clarify this further in the revised manuscript:

“Qualitative nutrient limitation regimes (including primary, co-, and serial limitations) for each experiment were designated on the basis of assessments originally made in the individual study publications, which was almost always via the authors of these studies undertaking a statistical test (e.g., t-test or ANOVA followed by various post-hoc tests) to establish whether there were significant differences in mean chlorophyll-a concentrations between nutrient amended bottles and non-amended control bottles.

Lines 471-472: same here, what does this mean, how was that done? The experimental duration was included but what if the experimental duration had an impact on the Chl a concentrations?

R3_8: We used Equation 1 to calculate net chlorophyll-a based growth rate. As indicated in the Methods text, the advantage of this over using the absolute change in chlorophyll-a concentration is that it removes the time dependence of the experiment result (longer experiments would be expected to produce a greater overall biomass change, as there is more time for the treatment related differences in phytoplankton growth to drive these changes) as well as that of the initial chlorophyll-a (i.e., a doubling in biomass would lead to a higher absolute biomass concentration change in a sample with higher initial chlorophyll-a).

Lines 478-480: Not clear how this correction for temperature could be applied for all regions. What about light or salinity-dependent effects on Chl a concentrations?

R3_9: The temperature correction could be applied to all experiments where the experimental temperatures were reported. Such experiments are typically set up to use saturating light levels, so there is no reason to expect there to be any difference in response magnitudes on this basis. The relatively minor variations in salinity which are encountered across open ocean systems are not known to have a significant physiological effect.

Lines 492-495: Unclear how this was done. The aim was also unclear; also how the data structure looked like. More details need to be provided here.

R3_10: The aim with this calculation was to rank dissolved nutrient concentration deficiencies (see Moore, 2016). We define nutrient deficiency as the lack of one nutrient with respect to another, taking into account the typical requirement of phytoplankton for a given nutrient. We could then use this ranking to see if the nutrient concentrations in seawater could predict which nutrient was found to be limiting in the bioassay experiment.

This was the procedure:

- 1) Obtain dissolved nutrient concentration data from the GEOTRACES IDP V2 dataset
- 2) Divide dissolved nutrient concentration value by the typical phytoplankton requirement (in units of moles nutrient per mole carbon – now explicitly defined; see below)
- 3) Numerically order the resultant values. The nutrient with the lowest value is the most deficient, and vice versa that with the highest value is most in excess.

We have now included a more complete description and also stated the ‘typical’ nutrient stoichiometry used (16 N: 1 P: 7.5×10^{-3} Fe: 2.8×10^{-3} Mn: 8.0×10^{-4} Zn : 1.9×10^{-4} Co).

Reference

Moore, C.M., 2016. Diagnosing oceanic nutrient deficiency. *Phil. Trans. Royal Soc. A* **374**, 20150290.

Results and discussion:

Line 83: Please explain how the 'nutrient limitation provinces' are refined. Wow did you calculate the percentage effects?

Adding the number of studies/data would be helpful to understand the impact of this information.

R3_11: Percentages were calculated from the number of experiments limited by a given nutrient (as reported by the individual studies, in turn informed by chlorophyll-a or primary production responses), divided by the total number of experiments. We have now added the numbers of experiments alongside the percentages.

Lines 154-165: Which data are without and with temperature correction and why is it relevant to show both? Is it necessary to show both for all figures?

R3_12: All available data were used in the 'with' and 'without' temperature correction comparison, noting that some studies did not report ambient experiment temperature which leads to a lower experimental number for the temperature-corrected statistics (n=765 for no temperature correction, n=680 for with temperature correction). We maintain that there is value in showing both the unnormalized and temperature normalized results as it is not possible to unequivocally ascribe the magnitudes of growth responses to temperature variability, as temperature and other ecosystem characteristics (e.g., chlorophyll-a concentrations) co-vary. We have now expanded upon this with some further statistical analysis presented in the Supplementary Information (new Supplementary Text 1), as well as making some modifications to the main text.

Lines 189-191: some statistical tests would help to strengthen this statement

R3_13: As noted in R3_12, we have now included a new Supplementary Text section (Supplementary Text 1) that includes details of further statistical analysis (summarized in Supplementary Tables 1 and 2). We now reference this within the revised manuscript text.

Lines 203-205: Impact of grazers, see comment above. This should be clear in order to use the data.

R3_14: As noted in R3_4 above, the impact of differential grazing between nutrient treatments cannot be unequivocally excluded, which is an assumption made in all nutrient-addition experiments assessing before and after incubation biomass changes (provided all grazers are not 100% excluded from all treatments, which is almost impossible in aquatic systems where grazers can be of similar sizes to phytoplankton). Also, as we note in R3_4, the typical high level of coherency in responses to nutrient supply (both in terms of replicates within a given treatment and between different nutrient combinations) suggests that within a given experiment grazing pressure is not a confounding factor on qualitative responses between treatments and not a major confounding factor on the magnitude of differential responses between treatments, but potentially needs to be considered when comparing experimental responses across different ecosystems with different plankton community structures.

Lines 221-222: more explanation is needed.

R3_15: We now reference the revised Methods section where details are provided on specifically how this calculation was conducted (also see R3_10).

Lines 238-240: more explanation is needed.

R3_16: We have now clarified this further in the revised text:

“Prediction of the results from the experimental dataset on the basis of independent dissolved nutrient deficiencies was also potentially complicated in a number of cases, where the concentrations of only a few nutrients were reported (typically N, P, and Fe).”

Lines 264-277: unclear why this is relevant

R3_17: This section provides support for quantitative prediction of net growth rates following nutrient supply using ambient nutrient concentration data (in addition to the qualitative predictions of which nutrient is limiting described in the previous section). This is relevant as it is of interest to be able to predict how much phytoplankton growth might occur following supply of different nutrient combinations.

Outlook:

This is a summary of the manuscript and some outlook is added. It is too long and has too much repetition of own results.

R3_18: This section has been edited to reduce summary components. The overall length however remains approximately the same, due to bringing in aspects requested from Reviewers 1 and 2.

Figures:

Figure 1: Very difficult to read. The legend should be better explained.

R3_19: The figure has now been amended to improve readability. The revised caption now reads:

“Figure 1. Global synthesis of nutrient limitation. (a) Experimental locations presented on a global map as coloured symbols. (b) Example experiment. Legend next to example experiment indicates the identities of (co-)limiting nutrient(s) in ‘a’. The central symbol colour(s) on the map indicate the primary limiting nutrient (i.e., adding this nutrient alone stimulated chlorophyll-a accumulation). Outer symbol colours (i.e., colours of the annulus) indicate serial limiting nutrient(s) (i.e., adding this nutrient in addition to the primary limiting nutrient(s) stimulated further growth than supplying the primary limiting nutrient(s) alone). Split colours for inner or outer symbol indicate nutrients that were co-limiting. Sequential levels of serial limitation are indicated by multiple layers of annuli, referencing to secondary limitation (inner annulus) and tertiary limitation (outer annulus). Co-limitation can either be at the primary (split central circle) or serial (split annulus) level. Mesoscale Fe enrichment experiments are shown as crosses. Background colours on map in ‘a’ indicate annual average surface nitrate concentrations. Regions of elevated soluble aerosol Fe deposition predicted by a model are highlighted⁷⁵.”

Figure 2: The number of studies would be useful information here. Especially for the multiple nutrient addition experiments.

R3_20: This information has been added in the titles of each sub-panel (and explained in the figure caption).

Figure 3: b&d: What does density mean here, this is not explained.

R3_21: This is the kernel density as calculated from the ‘density’ function in the R ‘stats’ package. This has now been included in the figure caption and revised Methods. The kernel density estimate is essentially a smoothed version of a histogram that is useful for displaying the distribution of continuous data (see e.g., Venables and Ripley, 2002).

Reference

Venables, W.N. and Ripley, B.D., 2002. *Modern applied statistics with S*. Springer Science & Business Media. pp. 126–133.

Figure 5: Legend: colors corresponding to Fig. 1 is not sufficient.

R3_22: This figure now has a stand-alone legend defining which colours refer to which nutrient.

Figure 8: Figure and data (incl. discussion), see above, not clear why it is relevant here.

R3_23: We think that comparison of our bioassay result findings with other proxies for nutrient limitation (in this case, measurements of nutrient stress) are highly relevant; specifically, our new compilation provides a means to assess the consistency of both approaches. Such approaches for assessing nutrient stress are expected to increase rapidly in deployment (for example via the Biogeosciences programme referred to; www.biogeosciences.org), such that they will likely rapidly overtake the coverage of nutrient addition experiments.

In the revised manuscript, more detail is given regarding the molecular biomarker datasets (noting that full details regarding data collection and data analysis are provided in the respective cited papers):

“Nutrient stress biomarker datasets. For comparison to the experimental dataset presented here, the *Prochlorococcus* nutrient stress biomarker datasets of Refs 57 and 61 were obtained from the supplemental materials of these respective papers. Nutrient stressors in Ref. 57 were defined here as (i) the presence of *P-II* indicating N stress, (ii) the presence of *idiA* as indicating Fe stress, (iii) the presence of both as indicating N-Fe co-stress. For the Ref. 61, all data come from their principal component analysis of nutrient stress genes.”

REVIEWERS' COMMENTS

Reviewer #1 (Remarks to the Author):

The authors did a good job addressing nearly all of my comments and suggestions. I support publication of the revised manuscript.

It will be an important and widely read work.

Reviewer #2 (Remarks to the Author):

The authors have satisfied my concerns. One visual note, because of the map projection use the polar regions are very difficult to discern, in particular the Ross Sea looks to be part of the Amundsen due to it being at the edge of the projection.